# BrainOOD: Out-of-distribution Generalizable Brain Network Analysis

**Jiaxing Xu**[1][*][†]**, Yongqiang Chen**[2][*]**, Xia Dong**[1]**, Mengcheng Lan**[3]**, Tiancheng Huang**[1]**,
Qingtian Bian**[1]**, James Cheng**[2]**, Yiping Ke**[1]

[1]College of Computing and Data Science, Nanyang Technological University
[2]Department of Computer Science and Engineering, The Chinese University of Hong Kong
[3]S-Lab, Nanyang Technological University
{jiaxing003, LANM0002, BIAN0027}@e.ntu.edu.sg;
{yqchen, jcheng}@cse.cuhk.edu.hk;
{xia.dong, tiancheng.huang, ypke}@ntu.edu.sg

## Abstract

In neuroscience, identifying distinct patterns linked to neurological disorders, such as Alzheimer's and Autism, is critical for early diagnosis and effective intervention. Graph Neural Networks (GNNs) have shown promising in analyzing brain networks, but there are two major challenges in using GNNs: (1) distribution shifts in multi-site brain network data, leading to poor Out-of-Distribution (OOD) generalization, and (2) limited interpretability in identifying key brain regions critical to neurological disorders. Existing graph OOD methods, while effective in other domains, struggle with the unique characteristics of brain networks. To bridge these gaps, we introduce *BrainOOD*, a novel framework tailored for brain networks that enhances GNNs' OOD generalization and interpretability. BrainOOD framework consists of a feature selector and a structure extractor, which incorporates various auxiliary losses including an improved Graph Information Bottleneck (GIB) objective to recover causal subgraphs. By aligning structure selection across brain networks and filtering noisy features, BrainOOD offers reliable interpretations of critical brain regions. Our approach outperforms 16 existing methods and improves generalization to OOD subjects by up to 8.5%. Case studies highlight the scientific validity of the patterns extracted, which aligns with the findings in known neuroscience literature. We also propose the first OOD brain network benchmark, which provides a foundation for future research in this field. Our code is available at https://github.com/AngusMonroe/BrainOOD.

## 1 Introduction

In neuroscience, a major goal is to identify distinct patterns linked to neurological disorders, such as Alzheimer's and Autism, by examining brain data of both healthy individuals and patients with these disorders (Poldrack et al., 2009). Among the neuroimaging techniques, resting-state functional magnetic resonance imaging (fMRI) is widely used to capture the functional connectivity between different brain regions (Worsley et al., 2002). fMRI can be modeled as brain networks, where each node represents a brain region, referred to as a region of interest (ROI), and each edge denotes the pairwise correlation between the blood-oxygen-level-dependent (BOLD) signals of two ROIs (Smith et al., 2011). These connections provide insights into how different brain regions co-activate or show correlated activities, offering a framework to study neurological systems through graph-based methods (Kawahara et al., 2017; Lanciano et al., 2020; Wang et al., 2023; Xu et al., 2024c).

The most prevalent brain network analysis model is based on Graph Neural Networks (GNNs), which have recently shown promising results (Li et al., 2019; 2021; Xu et al., 2024a). However, the application of GNN-based methods in brain network analysis poses two significant challenges.

---

[*]Equal contribution
[†]Corresponding author

First, brain network data are often collected from different sites, leading to *distribution shifts*, which severely degrade the performance of GNNs when generalizing to Out-of-Distribution (OOD) data during testing (Chen et al., 2022b; Xu et al., 2024b). Second, brain network analysis aims to uncover patterns that can facilitate early diagnosis and interventions for neurological disorders. This requires GNN models to possess strong *interpretability*, allowing them to identify key brain regions relevant to the concerned conditions.

Several interpretable GNN methods (Wu et al., 2022; Miao et al., 2022; Chen et al., 2024) have been proposed to address the OOD generalization problem. These methods assume that a causal subgraph contains the essential information for predictions, thus improving the robustness to distribution shifts. While this is effective in domains like molecular and social networks, such an approach struggles with brain network analysis. Unlike other graph-structured data, brain networks exhibit noise in both their structures and features. Existing methods primarily focus on extracting causal substructures, often overlooking the selection of critical node features, which limits their applicability to brain networks. Additionally, invariant subgraphs identified by these methods may not effectively interpret brain networks. As shown in Figure 1, invariant substructures involving different ROIs can reflect distinct functional implications, highlighting the need for a specialized approach. This leads to a key research question:

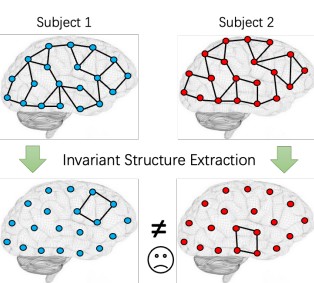

Figure 1: Same substructure in different brain regions may reflect distinct functional implications.

*How can one build an interpretable and OOD generalizable brain GNN?*

To tackle the aforementioned challenges, we propose the first benchmark dataset for evaluating OOD generalization performance in brain network analysis. Specifically, we go beyond the conventional usage of brain network datasets by creating a specific OOD benchmark scenario that simulates real-world conditions where models encounter data from unseen sites during testing. Building on this benchmark, we develop *BrainOOD*, which is a novel framework that enhances GNNs' representation power and enables the recovery of causal subgraphs using an improved Graph Information Bottleneck (GIB) objective (Wu et al., 2020). BrainOOD includes a feature selector and a structure extractor. The feature selector introduces a learnable masking process to selectively filter out noisy node features. A high-pass GNN with a reconstruction objective is incorporated to recover informative node features and learns high-quality representations to reveal causally interpretable brain regions. Additionally, we adopt a discrete sampling strategy for structure extraction. This ensures the identification of critical connections and enforces alignment across samples for consistent structure selection. Our contributions are summarized as follows:

- We introduce the first benchmark for evaluating OOD performance in brain network analysis. Our proposed benchmark is the first to systematically evaluate OOD generalization on brain network datasets with a focus on addressing site-specific variability, which is a critical challenge in clinical applications.
- We propose BrainOOD, a novel architecture that enhances OOD generalization in brain networks by selectively extracting node features and graph structures, while exploiting the inherent node alignment in brain networks.
- We evaluate BrainOOD against 16 existing methods and demonstrate its superior performance, improving generalization to OOD subjects by up to 8.5%.
- We present a case study to showcase the highly interpretable and scientifically meaningful patterns identified by BrainOOD, which align with the findings in neuroscience literature.

## 2 PRELIMINARIES

### 2.1 BRAIN NETWORKS CLASSIFICATION

We use the brain networks released by Xu et al. (2023). All preprocessed fMRI are parcellated by Schaefer atlas with 100 ROIs (Schaefer et al., 2018). For each subject, a brain network was constructed in the form of a connectivity matrix, $S$, where the nodes represent ROIs, and the edges

encode Pearson's correlation between the region-averaged BOLD signals of each pair of ROIs. Essentially, $S$ captures the functional relationships between different brain regions. To represent the brain network as a graph $G = (X, A)$, we define the feature matrix $X = S$, and the adjacency matrix $A$ as a sparsified version of $S$, retaining the top 20% of connections with the highest correlations. Notably, by using a consistent parcellation method, all brain networks share the same number of nodes $n = 100$, corresponding to the fixed set of ROIs.

Brain network classification aims to predict a subject's condition (e.g., autism diagnosis) based on his/her brain network. Given a dataset $\mathcal{D} = (\mathcal{G}, \mathcal{Y}) = \{(G, y_G)\}$, where $G \in \mathcal{G}$ represents a brain network and $y_G$ is its corresponding class label, the task is to learn a predictive function $f$: $\mathcal{G} \rightarrow \mathcal{Y}$ that maps brain networks to their respective labels. In this work, our objective of brain network classification is not only to accurately classify the networks in the training dataset but also to ensure that the learned function $f$ generalizes well to unseen or OOD brain networks, which may come from different sites with different feature distributions. In addition to the OOD generalizable predictions, we also aim to provide meaningful interpretations for the predictions by identifying a subgraph $G_C$ of the input brain network, offering insights into the functionalities of different ROIs. We summarize the notations used throughout the paper in Appendix A.

## 2.2 GRAPH NEURAL NETWORKS (GNNs)

GNNs have emerged as powerful tools for brain network analysis due to their ability to incorporate both node attributes and topological structures. Consider an input graph $G = (X, A)$, where $A$ is the adjacency matrix, which encodes connectivity information, and $X$ is the feature matrix containing attribute information for each node. The node set of $G$ is denoted as $\mathcal{V}_G$ and $|\mathcal{V}_G| = n$. The $l$-th layer of a GNN in the message-passing scheme (Xu et al., 2018) can be written as:

$$H_v^{(l)} = \text{AGG}^{(l-1)} \left( H_v^{(l-1)}, \text{MSG}^{(l-1)} \left( \left\{ H_u^{(l-1)} \right\}_{\forall u \in \mathcal{N}(v)} \right) \right), \tag{1}$$

where $H_v^{(l)} \in \mathbb{R}^d$ denotes the node representation at the $l$-th layer, where each node is represented by a $d$-dimensional vector. $\text{AGG}(\cdot)$ and $\text{MSG}(\cdot)$ are arbitrary differentiable aggregate and message functions (e.g., a multilayer perceptron (MLP) can be used as $\text{AGG}(\cdot)$ and a summation function as $\text{MSG}(\cdot)$). $\mathcal{N}(v)$ represents the neighbor node set of node $v \in \mathcal{V}_G$, and $H_v^{(0)} = X_v$ representing the raw features of node $v$.

In contrast to conventional message-passing GNNs, where information is aggregated from a node's neighbors, a high-pass graph neural network (HPGNN) emphasizes the differences between a node's features and the aggregated features of its neighbors. This approach is especially useful for capturing local variations in brain networks. The update rule for an HPGNN layer is:

$$H_v^{(l)} = H_v^{(l-1)} - \text{AGG}^{(l-1)} \left( \text{MSG}^{(l-1)} \left( \left\{ H_u^{(l-1)} \right\}_{\forall u \in \mathcal{N}(v)} \right) \right). \tag{2}$$

This operation enables the model to focus on deviations from local patterns, which may be critical in detecting abnormal or OOD graph substructures.

## 3 OUT-OF-DISTRIBUTION BENCHMARK IN BRAIN NETWORK ANALYSIS

### 3.1 DISTRIBUTION SHIFTS IN BRAIN NETWORK ANALYSIS

One of the primary goals in analyzing neurological disorders is to uncover disease-specific patterns that remain consistent across diverse populations. However, brain network datasets often exhibit distribution shifts (Xu et al., 2024b), where features common to specific sub-populations are mistakenly identified as disease-related, despite being unrelated to the disorder. This can result in models learning spurious connections that do not generalize across the broader population. For instance, large-scale brain network datasets like the Autism Brain Imaging Data Exchange (ABIDE) (Craddock et al., 2013) and the Alzheimer's Disease Neuroimaging Initiative (ADNI) (Dadi et al., 2019) are collected from multiple sites, such as various clinics or universities. Subjects from these different sites may introduce site-specific variability, such as differences in MRI scanner properties, or subject inclusion/exclusion criteria (Chan et al., 2022). Such factors contribute to site-specific biases,

where models inadvertently focus on site-related patterns rather than capturing population-invariant information about the disorders. The presence of this type of noise poses a significant challenge for model generalization, particularly in real-world medical applications, where deployment environments are rarely identical to training settings. Understanding and addressing these distribution shifts is crucial for improving the robustness and generalizability of brain network analysis models.

## 3.2 DATASET UNDER OOD SETTING

In medical applications, models are often trained on data collected from a limited number of sites but are expected to perform well across different, unseen sites during deployment. This scenario introduces OOD challenges, as variations between training and deployment sites can significantly degrade model performance. To investigate this OOD shift, we use two widely-studied, multi-site brain network datasets: ABIDE (Craddock et al., 2013), focused on Autism Spectrum Disorder (ASD), and ADNI (Dadi et al., 2019), centered around Alzheimer's Disease (AD). The statistics of these datasets are summarized in Table 1, and further detailed descriptions are provided in Appendix C.1. Both datasets were collected from multiple sites, with inherent inter-site variability in acquisition and processing methods. This variability provides an ideal testbed for evaluating model performance under OOD conditions. To simulate an OOD setting, we adopt a site-holdout strategy: each dataset is split into training, validation, and test sets in an 8:1:1 ratio. Importantly, the validation/test set is composed entirely of subjects from one specific site that were not present in the training set, making them OOD samples relative to the training data. This setup simulates the real-world scenario where a model trained on data from one set of sites is deployed in new, unseen environments. A detailed description of data split is included in Appendix C.2. For model evaluation, we use a consistent random seed across all experiments and perform 10-fold cross-validation. The average accuracy across folds is reported to ensure robustness in the results, allowing us to fairly compare models' generalization performance under OOD conditions.

Table 1: Statistics of Brain Network Datasets.

| Dataset | Condition | Subject# | Site# | Class# | Class Name |
|---------|-----------|----------|-------|--------|------------|
| ABIDE | Autism Spectrum Disorder | 1025 | 17 | 2 | {TC, ASD} |
| ADNI | Alzheimer's Disease | 1326 | 59 | 6 | {CN, SMC, EMCI, MCI, LMCI, AD} |

## 4 BRAINOOD

Brain networks differ from regular graph data in that the co-activity representations in brain networks can contain a lot of noise. Meanwhile, the interpretable biomarkers in brain network analysis are usually similar for the same target disorder. Therefore, it brings additional challenges in data modeling and objective design. In this section, we first demonstrate the failure of the existing GIB-based method and then propose several strategies to tackle the challenges.

## 4.1 INTERPRETABLE AND GENERALIZABLE BRAIN NETWORK ANALYSIS

In this work, our objective is to propose a robust GNN framework that can accurately predict the targets under distribution shifts. Meanwhile, we also aim to identify a subregion in brain networks to explain the target analysis results such as Autism Spectrum Disorder and Alzheimer's Disease, therefore, providing insights for future scientific discoveries.

Specifically, we adopt the Graph Information Bottleneck (GIB) framework (Wu et al., 2020; Miao et al., 2022; Chen et al., 2024), which can be formulated as follows:

$$\max_{G_C} I(G_C; y_G) - \beta I(G_C; G), \ G_C \sim g_\phi(G), \tag{3}$$

where $G_C$ encapsulates the causal information in $G$ that determines the target label $y_G$, $\beta \in [0, 1]$ is a trade-off hyperparameter, $g_\phi : \mathcal{G} \to \mathbb{G}(\mathcal{G})$ is the subgraph extractor parameterized by $\phi$, $\mathbb{G}(\mathcal{G})$ refers to the space of subgraphs for $G \in \mathcal{G}$, and $I(\cdot; \cdot)$ is the mutual information. Chen et al. (2022b); Miao et al. (2022); Chen et al. (2024) show that GIB can effectively solve for the desired causal subgraph $G_C^*$ in accordance with Eq. (3) under distribution shifts.

However, when applying GIB to brain networks, several new challenges arise: (a) **low informative features**, as the node features and connections refer to the co-occurrence of brain activities in differ-

ent ROIs; and (b) **unified interpretation**, as the interpretable ROIs for all subjects under the same condition should be similar.

Consequently, the expressiveness and the representational power of GNNs can be further limited when used to seek interpretable ROIs under the aforementioned constraints. The limited representational power of GNNs will further lead to suboptimal generalization and interpretations. More formally, we have the following theorem:

**Theorem 4.1.** *For a subgraph extractor $g_\phi$ that encodes the input graph $G$ into representation $\boldsymbol{H}$ to extract the desired subgraph $G_C^*$, if $g_\phi$ is limited in representation power, i.e., $I(G; \boldsymbol{H}) < H(G_C^*)$, where $H(\cdot)$ is the entropy of the underlying causal subgraph $G_C^*$, then solving for GIB objective (Eq. (3)) can not elicit $G_C^*$.*

The proof is given in Appendix B.1. Theorem 4.1 implies that it is essential to enhance the representation power of $g_\phi$ to effectively uncover the desired causal subgraph $G_C^*$. Consequently, we propose a new framework aimed at maximizing $I(G; \boldsymbol{H})$, while simultaneously incorporating an interpretation consistency regularization that ensures the structure of $G_C$ remains consistent across different samples.

The aforementioned gap motivates us to propose a novel graph OOD architecture, called *BrainOOD*, designed to offer both faithful interpretability and robust OOD generalizability. As shown in Figure 2, BrainOOD is composed of three main components: a feature selector, a structure extractor, and several auxiliary losses. These components work together to overcome the limitations of existing methods, ensuring that the model effectively captures discriminative connections while maintaining interpretability. The following sections provide a detailed description of each component and outline how they contribute to the promising performance and interpretability of BrainOOD.

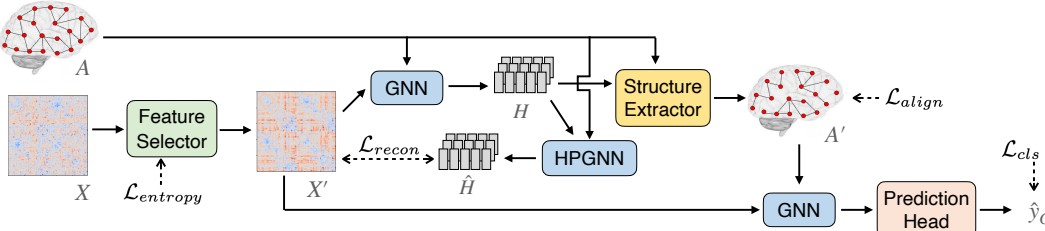

Figure 2: The framework of BrainOOD.

## 4.2 FEATURE SELECTION VIA RECONSTRUCTION

Brain network data can contain noise in specific ROIs, where GNNs may even amplify the noise due to the smoothing nature in message passing. This may further limit the extraction of useful information for the GIB objective. To address this, we introduce a learnable masking mechanism that filters out irrelevant connections and focuses on the most informative node features. This is followed by a reconstruction loss to identify key distinguishing features.

Given an input brain network $G = (\boldsymbol{X}, \boldsymbol{A})$, the masked features are obtained as:

$$\boldsymbol{X}' = \boldsymbol{X} \odot \boldsymbol{M}, \quad \boldsymbol{M} = \mathrm{Dropout}\left(\delta(\boldsymbol{W}_{mask}\boldsymbol{W}_{mask}^{\mathsf{T}})\right), \tag{4}$$

where $\odot$ is the Hadamard product, $\boldsymbol{W}_{mask} \in \mathbb{R}^{n \times d}$ is the learnable mask embedding, and $\delta$ is the sigmoid function. We employ an entropy loss as a sparsity constraint, to compel the model to prioritize the most informative connections and prevent a smooth mask. The entropy loss is formulated as follows:

$$\mathcal{L}_{entropy} = \frac{1}{n}\sum_{i=1}^{n} \mathrm{entropy}\left(\boldsymbol{M}(i, :)\right), \quad \mathrm{entropy}(\boldsymbol{p}) = -\sum_{j=1}^{n} \boldsymbol{p}_j \log(\boldsymbol{p}_j). \tag{5}$$

A GNN is subsequently employed to encode the brain network:

$$\boldsymbol{H} = \mathrm{GNN}(\boldsymbol{X}', \boldsymbol{A}). \tag{6}$$

It is well-known that GNN-based methods typically smooth node features across the graph, which can amplify noise in specific ROIs. To address this issue, we introduce a high-pass GNN to recover the input node features, guiding the model to learn the most informative features through a reconstruction loss:

$$\hat{\boldsymbol{X}} = \text{Tanh}(\hat{\boldsymbol{H}}\hat{\boldsymbol{H}}^{\mathsf{T}}) \odot \boldsymbol{M}, \quad \hat{\boldsymbol{H}} = \text{HPGNN}(\boldsymbol{H}, \boldsymbol{A}), \tag{7}$$

$$\mathcal{L}_{recon} = \text{MSE}(\hat{\boldsymbol{X}}, \boldsymbol{X}') = \frac{1}{n}\|\hat{\boldsymbol{X}} - \boldsymbol{X}'\|_F^2, \tag{8}$$

where $\|\cdot\|_F$ denotes the Frobenius norm. Herein, $\text{Tanh}(\cdot)$ serves to scale the range of the reconstructed features to align with the input connectivity matrix, while the self-multiplication operation is designed to ensure the output exhibits the symmetry property inherent in the connectivity matrix. This operation mimics the structure of the input data, making it easier for the model to capture meaningful patterns during reconstruction. By minimizing the Mean Square Error (MSE) between $\hat{\boldsymbol{X}}$ and $\boldsymbol{X}'$, the feature selector is trained to extract the most informative features $\boldsymbol{X}'$, ensuring the reconstruction is faithful to the input and improving the overall representation quality.

## 4.3 STRUCTURE EXTRACTION BY DISCRETE SAMPLING

Apart from node features, graph structure in brain networks also contains noise, which requires the model to extract critical substructures. When implementing the subgraph extractor $g_\phi$ in our improved GIB framework, we adopt the sampling strategy proposed by Chen et al. (2024). Specifically, an edge scorer is first applied to each edge in the input adjacency matrix based on the output of GNN encoder (Eq. (6)) as:

$$\alpha_{v,u} = \text{scorer}([\boldsymbol{H}_v | \boldsymbol{H}_u]), \quad \forall (v,u), \boldsymbol{A}_{v,u} = 1, \tag{9}$$

where $[\cdot|\cdot]$ is the concatenation function and $\text{scorer}(\cdot)$ can be arbitrary attention functions such as a simple MLP with Gumbel-softmax (Maddison et al., 2022). Thus the probability $\gamma_{v,u}$ of edge $(v,u)$ for sampling is defined as:

$$\gamma_{v,u} = \delta((\alpha_{v,u} + D)/\tau), \tag{10}$$

where $\tau$ is the temperature hyperparameter, $\delta(\cdot)$ is the sigmoid function, and $D = \log U - \log(1 - U)$, with $U \sim \text{Uniform}(0,1)$. To sample the discrete subgraph, we sample from the Bernoulli distributions on edges independently by $\boldsymbol{A}'_{v,u} \sim \text{Bern}(\gamma_{v,u})$.

Finally the generated desired causal subgraph $G_C^* = (\boldsymbol{X}', \boldsymbol{A}')$ is used to learn the node representation by $\boldsymbol{H}' = \text{GNN}(\boldsymbol{X}', \boldsymbol{A}')$. This sampling is done for $k$ times to do the independent prediction and obtain the logits $\hat{y}_i$. The final prediction is computed by the average of the $k$ simulated predictions: $\hat{y}_G = \frac{1}{k}\sum_{i=1}^{k}\hat{y}_i$.

## 4.4 LOSS FUNCTIONS

For brain networks, identifying specific ROIs or connections that correlate with neurological conditions is crucial for advancing our understanding of brain function and pathology. This task differs from traditional graph OOD methods, such as those proposed by Wu et al. (2022), Miao et al. (2022), and Chen et al. (2024), which focus on extracting invariant substructures across different graphs. While such methods work well for general graph analysis, they fall short in brain network analysis, where the same structural patterns involving distinct ROIs can reflect varying functional roles in brain activities (as shown in Figure 1).

In BrainOOD, we aim to discover key discriminative connections, rather than merely identifying invariant substructures. These connections may hold vital clues to understanding conditions like Alzheimer's and Autism by revealing the functional relationships between brain regions. To address this, we propose an alignment loss that encourages the structure extractor to consistently select the same connections across all brain networks within a batch:

$$\mathcal{L}_{align} = \frac{1}{n^2}\sum_{i=1}^{n}\sum_{j=1}^{n}\sigma'_{i,j}, \tag{11}$$

where $\sigma'$ is the standard deviation of all the $\boldsymbol{A}'$ in the batch. By applying this constraint, BrainOOD identifies the most informative connections, promoting both generalizability and interpretability in brain network analysis.

To incorporate domain knowledge and facilitate model convergence during optimization, we utilize 4 loss functions to guide the end-to-end training. Specifically, (1) a commonly-used cross-entropy loss (Cox, 1958) $\mathcal{L}_{cls} = \text{cross\_entropy}(\hat{y}_G, y_G)$ for graph classification; (2) an entropy loss $\mathcal{L}_{entropy}$ (Eq. (5)) for mask sparsification; (3) a reconstruction loss $\mathcal{L}_{recon}$ (Eq. (8)) to enforce the GNN to encode the most discriminative information; (4) an alignment loss $\mathcal{L}_{align}$ (Eq. (11)) to constrain node-identity awareness. The total loss is computed by:

$$\mathcal{L}_{total} = \mathcal{L}_{cls} + \lambda_1 * \mathcal{L}_{entropy} + \lambda_2 * \mathcal{L}_{recon} + \lambda_3 * \mathcal{L}_{align}, \tag{12}$$

where $\lambda_1$, $\lambda_2$ and $\lambda_3$ are trade-off hyperparameters.

## 5 EXPERIMENTAL RESULTS

### 5.1 BASELINE MODELS

We evaluate the proposed BrainOOD framework against a comprehensive set of baseline models, including **5 General OOD Methods**: ERM (Goyal, 2017), Deep Coral (Sun & Saenko, 2016), IRM (Arjovsky et al., 2019), GroupDRO (Sagawa et al., 2019) and VREx (Krueger et al., 2021); **4 Graph OOD Methods**: Mixup (Zhang et al., 2018), DIR (Wu et al., 2022), GSAT (Miao et al., 2022) and GMT (Chen et al., 2024) (All these graph OOD methods and BrainOOD are incorporated with GIN backbone for fair comparison); **2 conventional machine learning methods**: Support Vector Machine (SVM) and Logistic Regression (LR) Classifier from scikit-learn (Pedregosa et al., 2011), where these ML methods take the flattened upper-triangle connectivity matrix as vector input, instead of using the brain network; **3 General-Purpose GNNs**: GCN (Kipf & Welling, 2016), GIN (Xu et al., 2018) and GAT (Veličković et al., 2017); **4 Neural Networks Tailored for Brain Networks**: BrainNetCNN (Kawahara et al., 2017), BrainGNN (Li et al., 2021), ContrastPool (Xu et al., 2024a) and Contrasformer (Xu et al., 2024b). The detailed baseline description and implementation of these experiments are provided in Appendices D.1 and D.2, respectively.

Table 2: Results over 10-fold-CV (Average Accuracy ± Standard Deviation). The best result is highlighted in **bold** while the runner-up is highlighted in underline.

| OOD Model | ABIDE | | ADNI (5-class) | | ADNI (6-class) | |
|---|---|---|---|---|---|---|
| | ID | OOD | ID | OOD | ID | OOD |
| GCN | 63.69 ± 3.20 | 56.45 ± 5.52 | 59.95 ± 8.20 | 55.32 ± 10.23 | 61.01 ± 9.53 | 59.16 ± 11.75 |
| BrainNetCNN | **65.50** ± 4.77 | 60.38 ± 7.07 | 62.08 ± 6.81 | 55.02 ± 11.10 | 61.26 ± 7.26 | 56.73 ± 11.69 |
| ERM | 59.17 ± 6.99 | 56.73 ± 5.99 | 60.86 ± 9.17 | 60.81 ± 13.47 | 60.55 ± 10.11 | 59.32 ± 15.67 |
| Deep Coral | 60.40 ± 5.34 | 56.95 ± 5.94 | 62.22 ± 8.25 | 60.39 ± 15.51 | 58.25 ± 10.26 | 57.28 ± 13.55 |
| IRM | 58.73 ± 7.07 | 57.34 ± 8.74 | 61.94 ± 9.13 | 60.89 ± 11.32 | 62.36 ± 7.05 | 58.96 ± 13.42 |
| GroupDRO | 58.74 ± 8.43 | 58.83 ± 8.54 | 61.86 ± 8.34 | 57.34 ± 15.27 | 60.33 ± 7.74 | 54.33 ± 12.42 |
| VREx | 50.82 ± 2.11 | 52.08 ± 5.29 | 61.12 ± 6.71 | 55.64 ± 13.66 | 55.99 ± 9.45 | 50.07 ± 12.29 |
| Mixup | 62.06 ± 7.07 | 54.90 ± 7.71 | 62.82 ± 8.25 | 59.50 ± 12.81 | 60.51 ± 9.94 | 59.36 ± 13.73 |
| DIR | 59.77 ± 4.28 | 58.52 ± 9.61 | 65.83 ± 9.49 | 57.99 ± 14.82 | 58.48 ± 7.70 | 58.19 ± 16.09 |
| GSAT | 61.32 ± 6.37 | 57.57 ± 5.67 | 62.02 ± 8.77 | 60.27 ± 15.04 | 61.55 ± 10.55 | 60.79 ± 13.97 |
| GMT | 61.11 ± 6.30 | 59.73 ± 6.95 | 62.81 ± 6.54 | 60.93 ± 13.27 | 61.65 ± 11.37 | 58.13 ± 13.53 |
| BrainOOD | 64.07 ± 4.58 | **64.81** ± 9.01 | **66.09** ± 6.30 | **62.26** ± 15.83 | **65.71** ± 10.34 | **64.07** ± 14.99 |

### 5.2 MAIN RESULTS

We first compare BrainOOD with existing baselines in terms of in-domain (ID) and OOD classification accuracy. The results on 2 brain network datasets over 10-fold cross-validation (CV) are reported in Table 2. Apart from classifying ADNI into 6 classes, we also conduct experiment by a 5-class setting by merging MCI with LMCI to align with most AD diagnosis/prognosis studies in the literature. This is due to the MCI defined in ADNI 1 corresponding to LMCI in ANDI GO/2. Although non-OOD methods (GCN and BrainNetCNN) achieve good accuracy on ID set, they failed to generalize to OOD data. Most OOD algorithms have comparable performance with ERM, showing the difficulty of achieving invariant prediction in brain networks. While these graph OOD methods (Mixup, DIR, GSAT and GMT) apply well to graph topology, their failure to consider the unique characteristics of brain networks creates a performance bottleneck. On the contrary, our proposed BrainOOD leads to non-trivial improvements on both ID and OOD sets for all datasets. Especially, for the performance on OOD set, the improvement is up to 7.34% ((64.81% - 60.38%) / 60.38% =

Table 3: Results of more evaluation metrics over 10-fold-CV on the overall test set of ABIDE and ADNI datasets. The best result is highlighted in **bold** while the runner-up is highlighted in underline. For multiclass dataset of ADNI, all other metrics are the same as accuracy.

| Model | ABIDE | | | | | ADNI (5-class) | ADNI (6-class) |
| | Accuracy | Precision | Recall | micro-F1 | ROC-AUC | Accuracy | Accuracy |
|---|---|---|---|---|---|---|---|
| SVM | 61.56 ± 4.04 | 61.10 ± 3.57 | 63.02 ± 3.57 | 61.53 ± 7.28 | 60.89 ± 4.31 | 62.88 ± 4.75 | 61.24 ± 5.53 |
| LR | 61.23 ± 3.93 | 63.16 ± 2.89 | 62.72 ± 6.45 | 62.77 ± 3.81 | 61.32 ± 2.93 | 61.58 ± 4.52 | 61.05 ± 6.11 |
| GCN | 61.85 ± 4.39 | 60.13 ± 3.94 | 58.45 ± 10.67 | 58.88 ± 7.06 | 61.71 ± 4.59 | 61.92 ± 9.53 | 60.92 ± 4.13 |
| GIN | 56.49 ± 3.40 | 62.78 ± 12.71 | 28.52 ± 10.69 | 37.46 ± 7.56 | 55.22 ± 3.23 | 58.78 ± 9.53 | 59.29 ± 3.72 |
| GAT | 63.12 ± 4.72 | 61.50 ± 5.22 | 61.29 ± 6.75 | 61.20 ± 5.06 | 63.07 ± 4.67 | 60.94 ± 6.58 | 60.07 ± 5.34 |
| BrainNetCNN | 63.80 ± 4.44 | 62.38 ± 6.11 | 63.34 ± 8.11 | 62.35 ± 4.63 | **63.79** ± 4.32 | 59.77 ± 8.69 | 58.76 ± 3.09 |
| BrainGNN | 60.00 ± 3.96 | 58.94 ± 4.98 | 54.34 ± 7.30 | 56.23 ± 4.89 | 59.76 ± 3.93 | 62.08 ± 8.93 | 62.40 ± 4.44 |
| ContrastPool | 62.00 ± 2.97 | 56.02 ± 3.92 | **68.46** ± 12.60 | 62.84 ± 5.69 | 62.57 ± 3.93 | 61.22 ± 1.87 | 60.00 ± 5.54 |
| Contrasformer | 63.53 ± 3.03 | 60.73 ± 3.23 | 65.87 ± 6.30 | 63.01 ± 3.43 | 63.67 ± 3.02 | 63.52 ± 3.10 | 63.58 ± 6.06 |
| ERM | 60.00 ± 3.35 | 57.84 ± 5.15 | 57.43 ± 4.78 | 56.88 ± 4.99 | 57.47 ± 4.67 | 60.69 ± 4.32 | 59.60 ± 5.04 |
| Deep Coral | 59.71 ± 4.55 | 60.50 ± 5.08 | 59.46 ± 5.21 | 58.22 ± 5.54 | 58.97 ± 4.89 | 61.47 ± 3.42 | 57.74 ± 6.43 |
| IRM | 60.15 ± 4.97 | 61.34 ± 5.23 | 59.84 ± 4.61 | 58.81 ± 5.01 | 59.89 ± 4.56 | 61.16 ± 4.69 | 60.93 ± 4.96 |
| GroupDRO | 59.70 ± 2.89 | 60.91 ± 3.16 | 59.17 ± 3.59 | 58.24 ± 3.34 | 59.65 ± 3.13 | 59.84 ± 4.92 | 57.60 ± 4.16 |
| VREx | 57.47 ± 4.64 | 59.36 ± 4.92 | 53.82 ± 9.28 | 54.15 ± 7.44 | 57.06 ± 4.81 | 58.76 ± 3.79 | 54.11 ± 5.54 |
| Mixup | 60.30 ± 3.28 | 59.43 ± 5.00 | 58.16 ± 4.53 | 56.95 ± 3.98 | 58.23 ± 4.38 | 61.08 ± 3.27 | 60.00 ± 3.89 |
| DIR | 59.27 ± 6.41 | 60.45 ± 6.76 | 59.48 ± 6.84 | 58.22 ± 6.78 | 59.35 ± 6.76 | 62.16 ± 4.82 | 58.13 ± 6.29 |
| GSAT | 59.38 ± 3.54 | 59.73 ± 4.62 | 59.11 ± 4.28 | 58.15 ± 4.22 | 58.76 ± 4.01 | 60.92 ± 7.30 | 61.00 ± 6.02 |
| GMT | 60.95 ± 3.50 | 60.32 ± 3.29 | 59.96 ± 3.59 | 59.41 ± 3.69 | 59.81 ± 3.52 | 61.61 ± 6.44 | 60.00 ± 5.80 |
| BrainOOD (ours) | **63.95** ± 4.65 | **65.72** ± 5.24 | 63.37 ± 4.29 | **63.42** ± 4.86 | 63.52 ± 4.28 | **64.18** ± 5.48 | **64.80** ± 5.36 |

7.34% compared with BrainNetCNN). We further provide a deeper analysis for the performance distribution of graph OOD methods on each fold in Appendix E.1. BrainOOD consistently achieves top performance across multiple folds and maintains robustness in worst-case scenarios, demonstrating strong generalization capabilities to unseen sites.

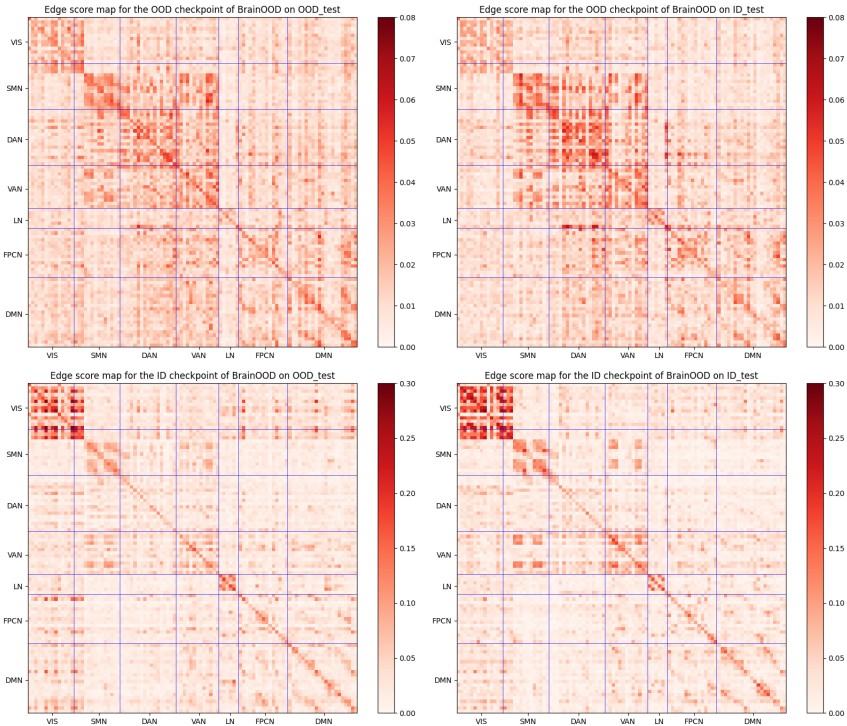

Figure 3: Edge score map visualization for ID/OOD checkpoints on ID/OOD test set of ABIDE dataset. VIS = visual network; SMN = somatomotor network; DAN = dorsal attention network; VAN = ventral attention network; LN = limbic network; FPCN = frontoparietal control network; DMN = default mode network.

To further compare BrainOOD with other general-purpose GNNs and neural networks specifically designed for brain networks, we report the results on the overall test sets in Table 3. Our proposed BrainOOD emerges as clear winner across both datasets. Interestingly, all existing OOD methods perform poorly, struggling even to surpass the simple GNN baselines. This suggests that current approaches to extracting invariant subgraphs are ineffective for brain networks and highlights the need for OOD algorithms that account for the unique characteristics of brain data. Notably, compared with the GIN backbone, incorporating our proposed OOD framework yields a significant 12.2% improvement, further verifying the effectiveness and necessity of BrainOOD in brain network analysis.

## 5.3 MODEL INTERPRETATION

In the domain of neurodegenerative disorder diagnosis, identifying significant ROIs and connections associated with predictions is critical, as these serve as potential biomarkers for diseases. For this study, we leverage edge scores from the structure extractor in BrainOOD to generate heat maps, providing interpretability of the model's predictions. These score maps are visualized using the Nilearn toolbox (Abraham et al., 2014). Figure 3 shows score maps for both ID and OOD checkpoints on the respective test sets from the ABIDE dataset, where higher scores signify stronger classification potential for ASD. We assessed the connections highlighted by our model in relation to Yeo's 7 networks (Yeo et al., 2011) that may be linked to the disorder. As shown in Figure 3, the score maps for the same checkpoint are consistent across both the ID and OOD test sets, suggesting that the model captures invariant patterns relevant to OOD subjects. Additionally, comparing different checkpoints on the same test sets reveals that both ID and OOD checkpoints identify common connections within key networks such as the somatomotor network (SMN), ventral attention network (VAN), and limbic network (LN), which are often associated with ASD (Hong et al., 2019; Farrant & Uddin, 2016). Interestingly, the score maps from the ID checkpoints tend to be sparser compared to those from OOD checkpoints. Furthermore, some connections are uniquely highlighted at different checkpoints, such as those within the visual network (VIS) for the ID checkpoint and within the dorsal attention network (DAN) for the OOD checkpoint.

To pinpoint the connections most significant for the causal subgraph, we selected the top 10 connections with the highest scores. Figure 4 highlights connections between the posterior, temporal occipital parietal regions in the ABIDE dataset, suggesting potential ASD-specific neural mechanisms. These regions align with prior research, which has identified them as critical areas in ASD studies (Ciaramidaro et al., 2018). Notably, these findings resonate with the discovery that adolescents with ASD exhibit hypo-activation in key visuoperceptual regions, particularly in the right hemisphere, as well as in affective and motivational face-processing areas (Scherf et al., 2015). A discussion of AD findings from the ADNI dataset is provided in Appendix E.3.

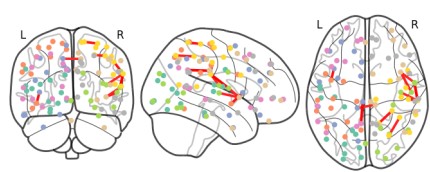

Figure 4: The visualization of the top 10 connections with the highest score on ABIDE OOD set.

## 5.4 ABLATION STUDY

To verify the effectiveness of our proposed components in BrainOOD, we test our design of the loss functions by disabling them one by one. The results are reported in Table 4, where "feat" and "adj" represent what we used as the feature matrix and adjacency matrix for the final prediction, respectively. We can observe that all of the auxiliary losses and components are effective in boosting the model performance. Besides, we find that the reconstruction loss and the alignment loss are important to ensure the ability of BrainOOD to generalize to the OOD set. This observation indicates the necessity of selecting information on both the feature and structure levels. We include the detailed hyperparameter sensitivity analysis in Appendix E.4.

## 6 RELATED WORKS

OOD or distribution shift is a longstanding problem in machine learning (Goyal, 2017; Zhang et al., 2018; Sagawa et al., 2019; Krueger et al., 2021). Most existing graph OOD methods aim to extract

Table 4: Ablation study on important components in BrainOOD on ABIDE dataset.

| feat | adj | $\mathcal{L}_{entorpy}$ | $\mathcal{L}_{recon}$ | $\mathcal{L}_{align}$ | ID acc | OOD acc | overall acc |
|------|-----|-----|-----|-----|--------|---------|-------------|
| $X'$ | $A'$ | | $\checkmark$ | $\checkmark$ | $63.92 \pm 4.13$ | $63.70 \pm 4.53$ | $63.12 \pm 2.50$ |
| $X'$ | $A'$ | $\checkmark$ | | $\checkmark$ | $62.82 \pm 4.19$ | $61.37 \pm 7.13$ | $61.85 \pm 4.53$ |
| $X'$ | $A'$ | $\checkmark$ | $\checkmark$ | | $63.26 \pm 3.44$ | $60.43 \pm 5.45$ | $61.85 \pm 2.83$ |
| $X$ | $A'$ | $\checkmark$ | $\checkmark$ | $\checkmark$ | $63.56 \pm 4.40$ | $62.26 \pm 5.68$ | $62.69 \pm 3.42$ |
| $X'$ | $A$ | $\checkmark$ | $\checkmark$ | $\checkmark$ | $63.71 \pm 5.97$ | $55.40 \pm 8.95$ | $60.10 \pm 3.47$ |
| $X'$ | $A'$ | $\checkmark$ | $\checkmark$ | $\checkmark$ | $\mathbf{64.07} \pm 4.58$ | $\mathbf{64.81} \pm 9.01$ | $\mathbf{63.95} \pm 4.65$ |

invariant subgraphs across all samples to enhance model generalization under distribution shifts. GIL (Li et al., 2022a) is a pioneering GNN-based model that identifies invariant subgraphs for graph classification tasks. It explores invariant graph representation learning in mixed latent environments without requiring labeled environments. DIR (Wu et al., 2022) introduces a causal inference approach to identify invariant causal parts through causal interventions. However, DIR involves a complex iterative process of breaking and assembling subgraphs during training. A more straightforward approach is GSAT (Miao et al., 2022), which is based on the information bottleneck principle and learns invariant subgraphs by reducing attention stochasticity. RGCL (Li et al., 2022b) combines invariant rationale discovery with contrastive learning to improve both generalization and interpretability. CIGA (Chen et al., 2022a) proposes an information-theoretic objective to extract invariant subgraphs, offering a theoretical guarantee for handling distribution shifts under different Structural Causal Models, which inspired a number of follow-up approaches (Chen et al., 2023; Yao et al., 2024). Similarly, GMT (Chen et al., 2024) focuses on extracting interpretable subgraphs by accurately approximating subgraph multilinear extensions, ensuring both interpretability and generalization under OOD conditions. A common finding across these invariant learning-based methods is the dependence on the diversity of environments. To address this, IGM (Jia et al., 2024) introduces a co-mixup strategy that combines environment and invariant mixups to generate diverse environments. These OOD methods that focus on extracting causal subgraphs work well in molecular and social networks but face challenges in brain network analysis due to the unique noise in both structures and features. These methods often overlook the selection of important node features, reducing their effectiveness for brain networks. Additionally, invariant subgraphs identified by these methods may not adequately capture the distinct functional implications of different brain regions, underscoring the need for a specialized approach. We include the discussion of more related works about brain network analysis with GNNs in Appendix F.

## 7 CONCLUSION

In this work, we introduced BrainOOD, a novel framework designed to tackle the dual challenges of OOD generalization and interpretability in brain network analysis. BrainOOD improves the representation power of GNNs through a feature selection process and a learnable masking mechanism, addressing the unique characteristics of brain networks by focusing on identifying critical connections rather than invariant substructures. The model's reconstruction loss further enhances its ability to reveal causally interpretable brain regions. Our extensive evaluations across 16 existing methods demonstrate that BrainOOD significantly outperforms both general-purpose and brain-specific GNNs, achieving up to an 8.5% improvement over existing graph OOD methods. Importantly, the model not only enhances OOD generalization but also extracts scientifically meaningful patterns that align with established knowledge in neuroscience. By presenting the first OOD benchmark dataset for brain network analysis, we provide a valuable resource for future research in improving both the generalizability and interpretability of models in this important domain of scientific research.

### ACKNOWLEDGMENTS

We thank the reviewers for their valuable comments. This research/project is supported by the Ministry of Education, Singapore under its MOE Academic Research Fund Tier 2 (STEM RIE2025 Award MOE-T2EP20220-0006) and Tier 1 (RG16/24). Any opinions, findings and conclusions or recommendations expressed in this material are those of the author(s) and do not reflect the views of the Ministry of Education, Singapore. YQ and JC are supported by Research Grants 8601116, 8601594, and 8601625 from the UGC of Hong Kong.

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
