# Appendix of BrainOOD

## CONTENTS

## A    NOTATION

Notation-wise, we use calligraphic letters to denote sets (e.g., $\mathcal{X}$), bold capital letters to denote matrices (e.g., $\boldsymbol{X}$), and strings with bold lowercase letters to represent vectors (e.g., $\boldsymbol{x}$). Subscripts and superscripts are used to distinguish between different variables or parameters, and lowercase letters denote scalars. We use $\boldsymbol{S}[i,:]$ and $\boldsymbol{S}[:,j]$ to denote the $i$-th row and $j$-th column of a matrix $\boldsymbol{S}$, respectively. Table 5 summarizes the notations used throughout the paper.

Table 5: Notation table

| Notation | Description |
|---|---|
| $\boldsymbol{S}$ | A connectivity matrix |
| $G$ | A brain network |
| $G_C$ | The causal subgraph for a brain network $G$ |
| $\boldsymbol{X}$ | The feature matrix of a brain network |
| $\boldsymbol{A}$ | The adjacency matrix of a brain network |
| $\mathcal{D}$ | Input dataset |
| $\mathcal{Y}$ | Input label set |
| $y_G$ | Label of brain network $G$ |
| $n$ | Number of nodes/ROIs |
| $\boldsymbol{H}_v$ | Node representation of $v$ |
| $d$ | Dimensionality of node representations |
| $\mathcal{G}$ | The graph space |
| $\mathcal{G}_C$ | The space of subgraphs with respect to the graphs from $\mathcal{G}$ |
| $\boldsymbol{W}_{mask}$ | Parameter matrices |
| $\boldsymbol{M}$ | The learnable mask |
| $\boldsymbol{X}'$ | The masked node feature matrix |
| $\hat{\boldsymbol{H}}$ | The recovered node representations |
| $\hat{\boldsymbol{X}}$ | The recovered node features with mask |
| $i, j, v, u$ | Index for matrix dimensions |
| $\boldsymbol{A}'$ | The sampled adjacency matrix |
| $\alpha_{v,u}$ | The score of edge $(v, u)$ |
| $\gamma_{v,u}$ | The sampling probability for edge $(v, u)$ |
| $\sigma'$ | The standard deviation matrix of all the $\boldsymbol{A}'$ in a batch |
| $g_\phi$ | The subgraph extractor with parameter $\phi$ to generate a subgraph $G_C$ to interpret brain network $G$ |
| $I(\cdot;\cdot)$ | Mutual information |
| $H(\cdot)$ | Entropy |
| $\hat{y}_G$ | The final prediction of brain network $G$ |

## B    THEORETICAL DISCUSSION AND PROOFS

### B.1    PROOF FOR THEOREM 4.1

**Theorem B.1** (Restatement of Theorem 4.1). *For a subgraph extractor $g_\phi$ that encodes the input graph $G$ into representation $\boldsymbol{H}$ to extract the desired subgraph $G_C^*$, if $g_\phi$ is limited in representation power, i.e., $I(G; \boldsymbol{H}) < H(G_C^*)$, where $H(\cdot)$ is the entropy of the underlying causal subgraph $G_C^*$, then solving for GIB objective:*

$$\max_{G_C} I(G_C; y_G) - \beta I(G_C; G), \ G_C \sim g_\phi(G), \tag{13}$$

*can not elicit $G_C^*$.*

*Proof.* Given the GIB objective, following previous works (Miao et al., 2022; Chen et al., 2024), we have:

$$
\begin{aligned}
I(G_C; y_G) - \beta I(G_C; G) &= I(y_G; G, G_C) - I(G; y_G | G_C) - \beta I(G_C; G) \\
&= I(y_G; G, G_C) - (1 - \beta) I(G; y_G | G_C) - \beta I(G; G_C, y_G) \\
&= (1 - \beta) I(y_G; G) - (1 - \beta) I(G; y_G | G_C) - \beta I(G; G_C | y_G).
\end{aligned}
\tag{14}
$$

Since $I(y_G; G)$ is fixed given the data generation process, maximizing Eq. (14) is equivalent to minimize $(1 - \beta)I(G; y_G|G_C) - \beta I(G; G_C|y_G)$. The minimizer is taken and only taken when $G_C = G_C^*$.

However, given the subgraph extractor $g_\phi$ that encodes the input graph $G$ into representation $\boldsymbol{H}$ to extract the desired subgraph $G_C^*$, we have a Markov chain $G_C^* \to G \to \boldsymbol{H} \to G_C$, from which we know that

$$I(G_C; G_C^*) \leq I(G; \boldsymbol{H}). \tag{15}$$

If $g_\phi$ is limited in representation is lower, i.e., $I(G; \boldsymbol{H}) < H(G_C^*)$, then it suffices to know that $I(G_C; G_C^*) < H(G_C^*)$, and $G_C \neq G_C^*$. $\square$

## C    MORE DETAILS ABOUT DATASETS

### C.1    DETAILED DATASET DESCRIPTION

The class-wise sample sizes are summarized in Table 6.

Table 6: The Class Distribution of the Brain Network Datasets we used

| Dataset | Gender (F/M) | Age (mean ± std) | Class | # Subjects |
|---|---|---|---|---|
| ABIDE | 152/873 | 16.5 ± 7.4 | Control | 537 |
| | | | ASD | 488 |
| ADNI | 728/599 | 74.6 ± 7.9 | CN | 819 |
| | | | SMC | 73 |
| | | | LMCI | 102 |
| | | | MCI | 179 |
| | | | EMCI | 89 |
| | | | AD | 65 |

**ABIDE** The ABIDE initiative supports the research on ASD by aggregating functional brain imaging data from laboratories worldwide. ASD is characterized by stereotyped behaviors, including irritability, hyperactivity, depression, and anxiety. Subjects in the dataset are classified into two groups: TC and individuals diagnosed with ASD.

**ADNI** The ADNI raw images used in this paper were obtained from the ADNI database (`adni.loni.usc.edu`). The ADNI was launched in 2003 as a public-private partnership, led by Principal Investigator Michael W. Weiner, MD. The primary goal of ADNI has been to test whether serial magnetic resonance imaging (MRI), PET, other biological markers, and clinical and neuropsychological assessment can be combined to measure the progression of mild cognitive impairment (MCI) and early AD. For up-to-date information, see `www.adni-info.org`. We include subjects from 6 different stages of AD, from cognitive normal (CN), significant memory concern (SMC), mild cognitive impairment (MCI), early MCI (EMCI), late MCI (LMCI) to AD.

### C.2    DETAILED DATA SPLITS UNDER OOD SETTING

Table 7 provides detailed information on the specific sites and the number of subjects used as OOD set in each fold. With such data split, the proportion of OOD subjects in the test set of each fold is in the range of [30%, 55%]. Subjects from the other sites are evenly assigned to each fold.

For the ABIDE dataset, given that the average number of subjects per site is approximately 60, we selected the smallest 10 sites as OOD sets across the 10 folds. This ensures that the test sets in all folds contain a mixture of both ID and OOD subjects, allowing for a robust evaluation of the model's generalization capabilities.

In contrast, for the ADNI dataset, where the number of sites is larger and the average number of subjects per site is only around 22, we selected the largest 10 sites as OOD sets across the 10 folds. This choice ensures that there are enough OOD subjects in the test set of each fold to reliably assess the model's performance under OOD conditions.

Table 7: The Site Chosen as OOD set in Each Fold of ABIDE and ADNI Datasets.

| Fold | ABIDE | | ADNI | |
|---|---|---|---|---|
| | Site Name | Subject# | SITEID | Subject# |
| 1 | SBL | 30 | 58 | 73 |
| 2 | OLIN | 36 | 59 | 62 |
| 3 | SDSU | 36 | 20 | 57 |
| 4 | CALTECH | 38 | 27 | 50 |
| 5 | STANFORD | 40 | 52 | 50 |
| 6 | TRINITY | 49 | 47 | 46 |
| 7 | KKI | 55 | 2 | 46 |
| 8 | YALE | 56 | 25 | 45 |
| 9 | MAX_MUN | 57 | 5 | 43 |
| 10 | PITT | 57 | 1 | 39 |

# D    MORE DETAILS ABOUT THE EXPERIMENTS

## D.1    BASELINE DESCRIPTIONS

- General OOD Methods.
  **ERM** (Goyal, 2017): Empirical Risk Minimization, which trains on the full dataset without specific domain adaptation.
  **Deep Coral** (Sun & Saenko, 2016): Minimizes the domain shift by aligning covariance matrices across domains.
  **IRM** (Arjovsky et al., 2019): Seeks to find invariant features across different environments by penalizing variations.
  **GroupDRO** (Sagawa et al., 2019): Tackles minority distributions by optimizing the worst-case group performance.
  **VREx** (Krueger et al., 2021): Reduces the risk variance across training environments to improve robustness.

- Graph OOD Methods.
  **Mixup** (Zhang et al., 2018): Trains the model on convex combinations of pairs of examples to enhance robustness.
  **DIR** (Wu et al., 2022): Selects causal subgraphs and conducts interventional augmentation to enhance OOD generalization.
  **GSAT** (Miao et al., 2022): Incorporates stochasticity in attention weights to filter task-irrelevant subgraphs while enhancing interpretability.
  **GMT** (Chen et al., 2024): Extracts interpretable subgraphs via approximation methods to achieve OOD generalization.

- General-Purpose GNNs.
  **GCN** (Kipf & Welling, 2016): A Graph Convolutional Network baseline with mean pooling.
  **GIN** (Xu et al., 2018): A Graph Isomorphism Network with sum pooling, which adjusts node importance using learnable parameters.
  **GAT** (Veličković et al., 2017): A Graph Attention Network, which applies attention mechanisms to learn node-to-neighbor importance weights.

- Neural Networks Tailored for Brain Networks.
  **BrainNetCNN** (Kawahara et al., 2017): A Convolutional Neural Network developed for connectome data.
  **BrainGNN** (Li et al., 2021): A GNN-based method that incorporates ROI-aware convolution layers for integrating fMRI data.
  **ContrastPool** (Xu et al., 2024a): A pooling method that clusters nodes and uses dual-attention mechanisms for domain-specific information.
  **Contrasformer** (Xu et al., 2024b): A transformer-based approach with contrastive constraints applied at both ROI and population levels.

## D.2 IMPLEMENTATION DETAILS

For all OOD methods, we use the same GNN architecture as graph encoders, following GSAT (Miao et al., 2022).We use 2-layer GIN (Xu et al., 2018) with Batch Normalization (Ioffe & Szegedy, 2015) as the backbone. The hidden dimension is set to 100 and the dropout ratio is set to 0.5. The pooling function is sum pooling. The settings of our experiments about OOD methods follow those in GOOD (Gui et al., 2022). The whole network is trained in an end-to-end manner using the Adam optimizer (Kingma & Ba, 2014) with a learning rate of 1e-3 and a batch size of 64 for all OOD models at all datasets. All OOD models are trained for 100 epochs. The final model is selected according to the best validation classification performance on ID and OOD sets, respectively. We report the mean and standard deviation of 10 folds to evaluate how these models can generalize to the unseen OOD sites. All the codes were implemented using PyTorch (Paszke et al., 2017) and PyTorch Geometric (Fey & Lenssen, 2019) packages. The optimized hyperparameters for BrainOOD are reported in Table 8.

Table 8: The optimized hyperparameters for BrainOOD.

|  | ABIDE | ADNI |
| --- | --- | --- |
| feature dropout | 0.2 | 0.2 |
| $\lambda_1$ | 0.01 | 0.01 |
| $\lambda_2$ | 0.1 | 10 |
| $\lambda_3$ | 0.5 | 0.1 |
| $k$ | 5 | 3 |

The experiments of general-purposed GNNs and models tailored for brain networks based on the framework used in ContrastPool (Xu et al., 2024a). The learning rate and batch size are using author-recommended values for fair comparison. The maximum number of training epochs is set to 1000. We use the early stopping criterion, i.e., we stop the training once there is no further improvement on the validation loss during 25 epochs. The whole network is trained in an end-to-end manner using the Adam optimizer (Kingma & Ba, 2014) with.

All experiments were conducted on a Linux server with an Intel(R) Core(TM) i9-10940X CPU (3.30GHz), a GeForce GTX 3090 GPU, and a 125GB RAM.

## E MORE EXPERIMENTAL RESULTS

### E.1 IN-DEPTH ANALYSIS FOR THE PERFORMANCE ON DIFFERENT SITES

We also conducted a detailed evaluation of the OOD set in each fold, which reveals how well the models generalize to unseen sites. Figure 5 presents a comparison of BrainOOD against four other graph OOD methods. The trends across different folds on the two datasets are consistent, and we observe large variances for accuracy across different folds, especially on ADNI dataset. This indicates that some sites are significantly different from others, making it difficult for models to generalize effectively to these sites.

On the ABIDE dataset, BrainOOD achieves the best results on 6 out of 10 folds and secures the second-best performance on 2 other folds. BrainOOD surpasses the runner-up model up to 10% (on fold 2). Similarly, on the ADNI dataset, BrainOOD also ranks first on 6 out of 10 folds and second-best on 2 additional folds. BrainOOD surpasses the runner-up model up to 6% (on fold 8). Notably, BrainOOD never ranks as the worst-performing model across all folds of both datasets. The worst performance for BrainOOD is still the best compared to the worst one of other models on all folds of these datasets.

These results demonstrate that BrainOOD not only has strong generalization capabilities but also exhibits robustness in its performance across multiple unseen sites, making it a reliable choice for OOD scenarios in brain network analysis.

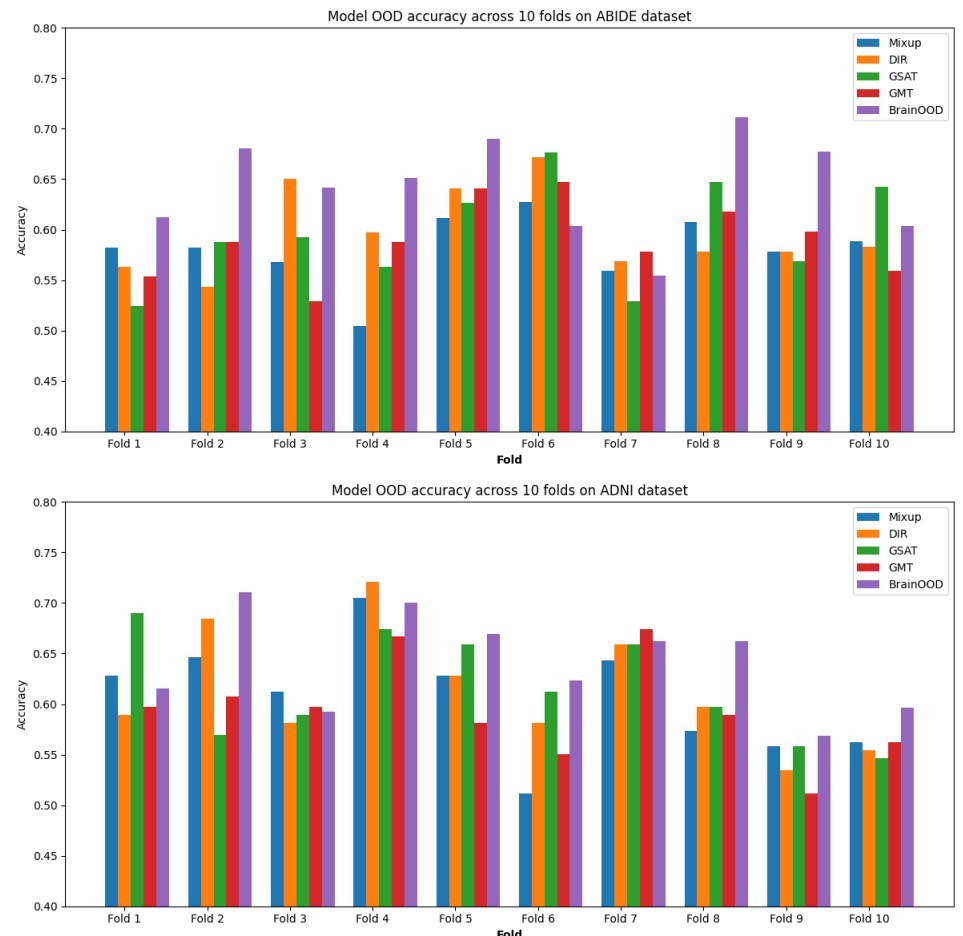

Figure 5: Comparison with graph OOD methods in terms of test OOD accuracy across 10 folds on ABIDE and ADNI datasets.

## E.2 EXPERIMENTAL RESULTS WITH OTHER BACKBONE

To verify the adaptability of the BrainOOD framework to different GNN backbones, we conducted experiments by integrating various graph OOD methods with GCN backbones. The results, presented in Table 9, demonstrate that existing OOD methods fail to improve performance when combined with the GCN backbone, emphasizing the necessity of designing OOD algorithms specifically tailored for brain networks.

In contrast, integrating BrainOOD with the GCN backbone results in a notable improvement, achieving a 6.3% increase in overall accuracy. This significant gain highlights the effectiveness of BrainOOD in enhancing the generalization capabilities of GNN models for brain network analysis, even when applied to general-purpose backbones like GCN.

Table 9: Results of graph OOD methods with GCN backbone. The best result is highlighted in **bold**.

| Model | ABIDE | | | ADNI (6-class) | | |
|---|---|---|---|---|---|---|
| | ID acc | OOD acc | Overall acc | ID acc | OOD acc | Overall acc |
| GCN | - | - | 61.85 ± 4.39 | - | - | 60.92 ± 4.13 |
| Mixup | 60.78 ± 5.01 | 58.06 ± 6.06 | 59.52 ± 3.93 | 59.34 ± 7.52 | 60.01 ± 13.65 | 59.69 ± 5.37 |
| DIR | 60.66 ± 6.53 | 57.81 ± 5.56 | 59.76 ± 2.69 | 60.71 ± 10.04 | 60.20 ± 14.18 | 60.23 ± 5.05 |
| GSAT | 62.73 ± 4.47 | 59.12 ± 6.17 | 61.27 ± 2.03 | 58.67 ± 10.02 | 57.99 ± 15.37 | 57.89 ± 7.19 |
| GMT | 63.38 ± 5.23 | 58.14 ± 7.41 | 61.56 ± 4.05 | 60.34 ± 11.00 | 56.31 ± 11.28 | 58.68 ± 6.92 |
| BrainOOD | **64.91** ± 4.23 | **62.85** ± 6.88 | **63.34** ± 2.77 | **66.54** ± 11.51 | **62.05** ± 14.50 | **64.10** ± 5.16 |

## E.3 MODEL INTERPRETATION WITH ADNI

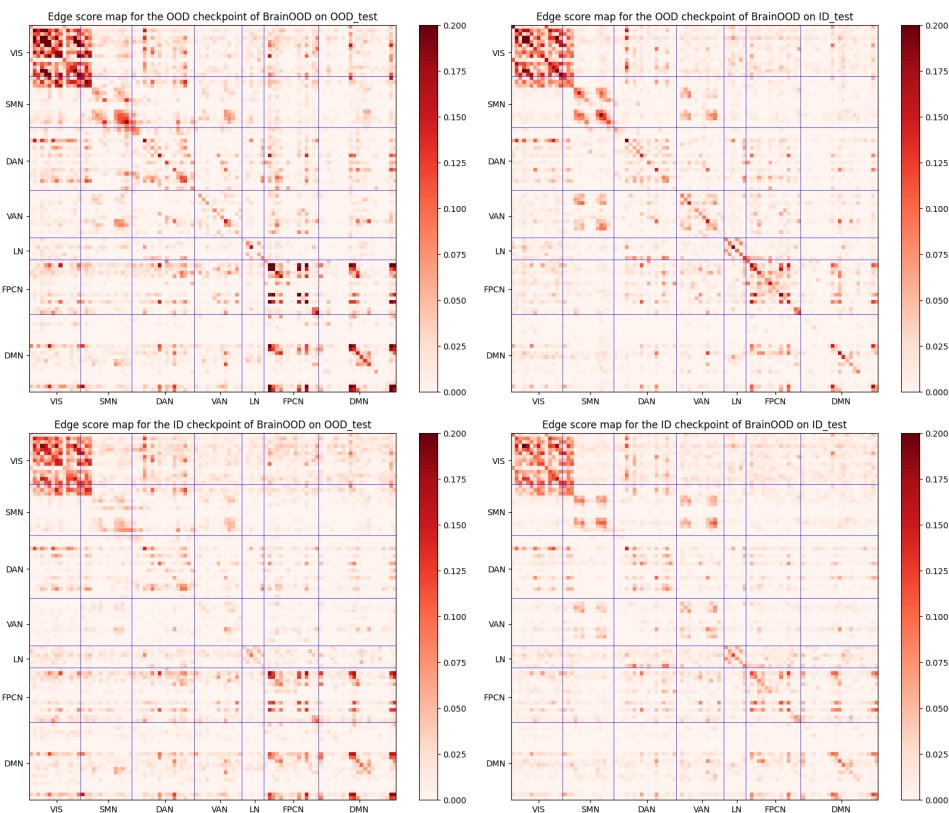

Figure 6: Edge score map visualization for ADNI dataset. VIS = visual network; SMN = somato-motor network; DAN = dorsal attention network; VAN = ventral attention network; LN = limbic network; FPCN = frontoparietal control network; DMN = default mode network.

In the ADNI dataset, we observed similar consistency between score maps for both ID and OOD test sets when evaluated using the same checkpoint, as illustrated in Figure 3. This consistency once again highlights BrainOOD's ability to capture invariant patterns from OOD subjects. When comparing different checkpoints on the same test sets, both ID and OOD checkpoints identify common connections within VIS and frontoparietal control network (FPCN), both of which are recognized as important connectivity regions in AD research (Jiang et al., 2020; Boyle et al., 2024). Additionally, some connections, such as those within SMN, are uniquely highlighted in the OOD checkpoint, emphasizing the variations that may arise between the different test environments.

For the most significant connections in the causal subgraph of ADNI, we selected the top 10 connections with the highest scores, as shown in Figure 7. These highlighted connections across the left and right hemispheres, particularly between the lateral prefrontal cortex and medial posterior prefrontal cortex, suggest potential AD-specific neural mechanisms. Previous studies have identified these regions as critical in AD progression (Venneri et al., 2008; McGeown et al., 2009). Notably, research also indicates that interhemispheric connectivity, particularly involving the corpus callosum, plays a crucial role in AD (Wang et al., 2015), further validating our model's interpretability in identifying AD-relevant neural patterns.

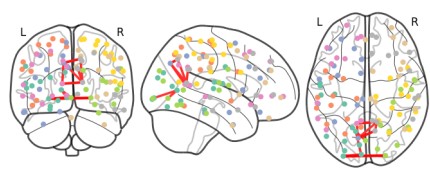

Figure 7: The visualization of the top 10 connections with the highest score on ADNI OOD set.

### E.4 HYPERPARAMETER ANALYSIS

In this section, we study the sensitivity of three trade-off hyperparameters in Eq. (12) and the sampling number $k$. All experiments are conducted on the ABIDE dataset. We tune the value of $\lambda_1$ from $\{0.001, 0.01, 0.1\}$, $\lambda_2$ from $\{0.01, 0.1, 1.0\}$, $\lambda_3$ from $\{0.1, 0.5, 1.0\}$, and $k$ from $\{1, 3, 5, 10, 20\}$. The results presented in Table 10 show that our model performs the best when $\lambda_1 = 0.01$, $\lambda_2 = 0.1$, $\lambda_3 = 0.5$, and $k = 5$. We can exhibit that the influence of $\lambda_1$ and $\lambda_2$ is larger than $\lambda_3$, which implies the importance of introducing feature selection with a suitable trade-off.

Table 10: The hyperparameter sensitivity analysis for BrainOOD on ABIDE dataset.

| $k$ | $\lambda_1$ | $\lambda_2$ | $\lambda_3$ | overall acc |
|---|---|---|---|---|
| 1 | 0.01 | 0.1 | 0.5 | $61.95 \pm 4.54$ |
| 3 | 0.01 | 0.1 | 0.5 | $61.37 \pm 3.38$ |
| 5 | 0.001 | 0.1 | 0.1 | $61.31 \pm 5.26$ |
| 5 | 0.001 | 0.1 | 0.5 | $62.19 \pm 3.45$ |
| 5 | 0.001 | 0.1 | 1.0 | $61.98 \pm 5.56$ |
| 5 | 0.01 | 0.01 | 0.1 | $61.71 \pm 3.49$ |
| 5 | 0.01 | 0.01 | 0.5 | $62.52 \pm 4.15$ |
| 5 | 0.01 | 0.01 | 1.0 | $61.71 \pm 3.49$ |
| 5 | 0.01 | 0.1 | 0.1 | $62.98 \pm 3.57$ |
| 5 | 0.01 | 0.1 | 0.5 | $\mathbf{63.95} \pm 4.65$ |
| 5 | 0.01 | 0.1 | 1.0 | $62.72 \pm 4.00$ |
| 5 | 0.01 | 1.0 | 0.1 | $62.05 \pm 5.14$ |
| 5 | 0.01 | 1.0 | 0.5 | $61.15 \pm 2.84$ |
| 5 | 0.01 | 1.0 | 1.0 | $60.59 \pm 5.24$ |
| 5 | 0.1 | 0.1 | 0.1 | $61.46 \pm 4.41$ |
| 5 | 0.1 | 0.1 | 0.5 | $61.66 \pm 3.65$ |
| 5 | 0.1 | 0.1 | 1.0 | $62.00 \pm 4.50$ |
| 10 | 0.01 | 0.1 | 0.5 | $62.90 \pm 4.67$ |
| 20 | 0.01 | 0.1 | 0.5 | $61.59 \pm 3.57$ |

## F MORE RELATED WORKS ABOUT BRAIN NETWORK ANALYSIS WITH GNNS

In recent years, several GNN-based methods have been proposed for brain network analysis. Ktena et al. (2017) leverages graph convolutional networks (GCNs) for learning similarities between each pair of graphs (subjects). BrainNetCNN (Kawahara et al., 2017) proposes edge-to-edge, edge-to-node and node-to-graph convolutional filters to leverage the topological information of brain networks in the neural network. PRGNN (Li et al., 2020) proposes a graph pooling method with group-level regularization to guarantee group-level consistency. BrainGNN (Li et al., 2021) proposes an ROI-selection pooling to highlight salient ROIs for each individual. MG2G (Xu et al., 2021) is a two-stage approach. The first stage learns node representations through a self-supervised link prediction task. The second stage employs the learned representations to train a classifier for predicting Alzheimer's disease progression. LG-GNN (Zhang et al., 2022) incorporates local ROI-GNN and global subject-GNN guided by non-imaging data, such as gender, age, and acquisition site. Some more recent works (Xu et al., 2024a;b) introduce a contrast graph to highlight the difference between groups and thus improve the model's generalization ability. Despite these advancements, addressing the OOD challenge in brain network analysis remains largely unexplored. Furthermore, while data harmonization methods (Guan et al., 2021; Wang et al., 2022) and domain adaptation methods (Lei et al., 2023; Liu et al., 2023) have been widely applied in study generalizing brain network models to other sites. However, these methods typically rely on learning a mapping from a source to a target domain, assuming the availability of the target domain distribution during training. In contrast, our study addresses the OOD generalization setting, where target domain data is entirely unseen during training. This stricter constraint represents a more challenging and realistic scenario, particularly in clinical applications where models must generalize to previously unseen sites without retraining. As a result, domain adaptation methods may be less effective in this context. Our work, therefore,

pioneers the evaluation of brain network classification under an OOD generalization framework, emphasizing the need for new OOD algorithms specifically designed for brain networks.