# OpenReview forum: "BrainOOD: Out-of-distribution Generalizable Brain Network Analysis"
_ICLR.cc/2025/Conference — ICLR 2025 Poster_

### Official Review · Reviewer_GyEy · 2024-11-02

**Soundness:** 3
**Presentation:** 2
**Contribution:** 2
**Rating:** 5
**Confidence:** 4

**Summary:**

This paper presents BrainOOD, a novel GNN framework tailored for brain functional network analysis, which consists of feature selector and causal subgraph extractor for brain functional network to enhance the generalization to out-of-distribution dataset. The proposed framework has been evaluated on two multi-site datasets and demonstrated improved classification performance.

**Strengths:**

It is novel to simultaneously identify informative features and extract causal subgraph for brain functional network based prediction.

**Weaknesses:**

1. Several descriptions are not clear. Please refer to the Questions section for details.
2. The classification setting (6-class) on the ADNI dataset. It is confusing to have three classes related to MCI (MCI, EMCI, and LMCI), which affects the evaluation results. EMCI and LMCI are used in ANDI GO/2, while MCI used in ADNI 1 is deemed LMCI. A 5-class (CN, SMC, EMCI, LMCI, AD) setting is more reasonable.

**Questions:**

1. For the adjacency matrix, were the top 20% connections identified based on correlation magnitude (including both positive and negative correlation)?
2. The classification setting (6-class) on the ADNI dataset. It is confusing to have three classes related to MCI (MCI, EMCI, and LMCI), which may affect the evaluation results. EMCI and LMCI are used in ANDI GO/2, while MCI used in ADNI 1 is deemed LMCI. A 5-class (CN, SMC, EMCI, LMCI, AD) setting is more reasonable.
3. It would be helpful to add more description about how the reconstruction loss can help select informative features.
4. It is not clear how in-domain testing was performed.
5. What are the differences between the 10-fold-CV and the overall test in Tabel 2 and 3?
6. For evaluation, it is better to add some conventional ML methods (e.g., SVM) as baseline.
7. What does the ID and OOD checkpoints mean in Fig.3? The edge score seems quite low (max value around 0.08, Fig.3 top left), how many edges were generally included in the extracted sub-graph?
8. There are several other parameters in the framework (e.g., temperature in eq.11, number of sampling k for the final prediction). How do they affect the performance?

---

> ### Author Response · Authors · 2024-11-21
> **Response to Reviewer GyEy [1/2]**
>
> **[W1, Q2. Experiments for ADNI with 5 Classes]**   Thank you for pointing this out. To maintain consistency with prior studies, we initially adopted the 6-class setting. However, we acknowledge that distinguishing MCI, EMCI, and LMCI introduces complexity. In response to your suggestion, we conducted experiments using a 5-class setting (CN, SMC, EMCI, LMCI, AD) by merging MCI with LMCI. As shown in the following table, our proposed BrainOOD consistently achieved the best results on both ID and OOD sets under this revised setting. We have included the detailed results and discussion in Appendix E.1 of our revision.
>
> |     OOD model     |     ID acc          |     OOD acc          |     acc             |
> |-------------------|---------------------|----------------------|---------------------|
> |     ERM           |     60.86 ± 9.17    |     60.81 ± 13.47    |     60.69 ± 4.32    |
> |     Deep Coral    |     62.22 ± 8.25    |     60.39 ± 15.51    |     61.47 ± 3.42    |
> |     Mixup         |     62.82 ± 8.25    |     59.50 ± 12.81    |     61.08 ± 3.27    |
> |     IRM           |     61.94 ± 9.13    |     60.89 ± 11.32    |     61.16 ± 4.69    |
> |     GroupDRO      |     61.86 ± 8.34    |     57.34 ± 15.27    |     59.84 ± 4.92    |
> |     VREx          |     61.12 ± 6.71    |     55.64 ± 13.66    |     58.76 ± 3.79    |
> |     DIR           |     65.83 ± 9.49    |     57.99 ± 14.82    |     62.16 ± 4.82    |
> |     GSAT          |     62.02 ± 8.77    |     60.27 ± 15.04    |     60.92 ± 7.30    |
> |     GMT           |     62.81 ± 6.54    |     60.93 ± 13.27    |     61.61 ± 6.44    |
> |     BrainOOD      |    **66.09** ± 6.30    |    **62.26** ± 15.83    |     **64.18** ± 5.48    |
>
>
>
> **[Q1. Thresholding Detail]**  The adjacency matrix was constructed by thresholding the connectivity matrix to retain the top 20% of values based on correlation magnitude. To ensure compatibility with GNNs, which do not handle negative edges effectively, we retained only positive connections. This approach follows the dataset paper [1] and is a common practice in GNN-based brain network analysis [2].
>
>
> **[Q3. Reconstruction Loss for Feature Selection.]** The reconstruction loss plays a crucial role in guiding the model to focus on informative features. By minimizing the discrepancy between the original and reconstructed features, the loss encourages the model to mitigate the limited information recovery (Theorem 4.1), and hence to retain only the most significant features while filtering out noise. This mechanism ensures that the extracted features align with the functional relevance of the brain network.
>
> **[Q4. In-Domain Testing.]** In-domain (ID) testing was conducted using data from sites that the model has seen in the training set. The ID performance was evaluated using a separate held-out portion of the training data to ensure that the results reflect the model's ability to generalize to data from known distributions.
>
> **[Q5, Q7. ID and OOD checkpoints.]** Thank you for highlighting potential ambiguities. Both Tables 2 and 3 follow the 10-fold CV setting. We revised the caption of Table 3 to avoid misunderstanding.
> - Table 2: ID and OOD accuracies are reported using separate models selected via the validation set for ID and OOD data, respectively.
> - Table 3: Overall accuracy combines ID and OOD results through a weighted average for a fair comparison with non-OOD methods.
> In Figure 3, the ID and OOD checkpoints refer to the models selected for ID and OOD evaluation. The visualized edge scores (e.g., max value ~0.08 in Fig. 3) represent $W_{Mask}W_{Mask}^T$ without sigmoid from Eq. (4). Post-sigmoid, even edges with scores as low as 0.08 exhibit strong relative importance. Regarding the subgraph, edge sampling is performed $k$ times based on learned probabilities for independent predictions. This dynamic sampling approach makes it difficult to define the exact number of edges in the subgraph.

---

> > ### Comment · Reviewer_GyEy · 2024-11-25
> >
> > Thanks for your responses and the additional experimental results, which address most of my concerns. For the experiments on ADNI dataset, I would suggest putting the 5-class results in the main text, as that setting is consistent with most AD diagnosis/prognosis studies in the literature.

---

> > > ### Author Response · Authors · 2024-11-27
> > >
> > > Dear Reviewer GyEy,
> > >
> > > We sincerely appreciate your valuable feedback and the time you’ve taken to review our work. Based on your suggestion, we have updated all experimental results on the ADNI dataset (**in the main text**) to follow the 5-class classification setting (CN, SMC, EMCI, LMCI, AD). We believe this adjustment better aligns with the existing AD literature and addresses your concern regarding the classification setting.

---

> ### Author Response · Authors · 2024-11-21
> **Response to Reviewer GyEy [2/2]**
>
> **[Q6. Conventional ML Methods as Baselines.]** Including traditional machine learning models as baselines is a good suggestion. We conduct experiments with SVM and LR by following the setting in [1], where these ML methods take the flattened upper-triangle connectivity matrix as vector input, instead of using the brain network. We update the results in Table 3 of our revision, in which our proposed BrainOOD still achieves the best performance.
>
> |              |     ABIDE           |                     |                     |                     |                     |     ADNI            |
> |--------------|---------------------|---------------------|---------------------|---------------------|---------------------|---------------------|
> |     model    |     acc             |     precision       |     recall          |     F1              |     ROC-AUC         |     acc             |
> |     SVM      |     61.56 ± 4.04    |     61.10 ± 3.57    |     63.02 ± 3.57    |     61.53 ± 7.28    |     60.89 ± 4.31    |     62.88 ± 4.75    |
> |     LR       |     61.23 ± 3.93    |     63.16 ± 2.89    |     62.72 ± 6.45    |     62.77 ± 3.81    |     61.32 ± 2.93    |     61.58 ± 4.52    |
>
>
> **[Q8. How other parameters affect the performance.]** Thank you for your comment. For the temperature hyperparameter in Eq. (10), we just set it as $\tau=1.0$ by following the default setting in the previous work [3]. This setting works consistently well, allowing us to use these default values without further tuning when adapting to new datasets. For the number of sampling $k$, we conduct additional experiments by tuning it from {1, 3, 5, 10, 20}. We added the results and discussion in Appendix E.5 of our revision.
>
> |     k     |     acc             |
> |-----------|---------------------|
> |     1     |     61.95 ± 4.54    |
> |     3     |     61.37 ± 3.38    |
> |     5     |     **63.95** ± 4.65    |
> |     10    |     62.90 ± 4.67    |
> |     20    |     61.59 ± 3.57    |
>
>
>
> [1] Data-driven network neuroscience: On data collection and benchmark. NeurIPS 2023
>
> [2] Braingnn: Interpretable brain graph neural network for fmri analysis. MIA 2021
>
> [3] How interpretable are interpretable graph neural networks? ICML 2024

---

### Official Review · Reviewer_G3Bt · 2024-11-04

**Soundness:** 3
**Presentation:** 3
**Contribution:** 2
**Rating:** 5
**Confidence:** 4

**Summary:**

The paper presents BrainOOD, a framework designed to address the challenges of Out-of-Distribution (OOD) generalization in brain network analysis. Specifically, BrainOOD aims to enhance the performance and interpretability of Graph Neural Networks (GNNs) in diagnosing Alzheimer’s Disease (AD) and Autism Spectrum Disorder (ASD). The method incorporates a feature selector, structure extractor, and auxiliary losses, leveraging the Graph Information Bottleneck (GIB) framework to recover causal subgraphs. Through extensive experiments on those datasets, the framework demonstrates competitive performance, outperforming baseline models in OOD settings.

**Strengths:**

- The paper addresses a critical gap in brain network analysis by focusing on OOD generalization and interpretability, which are essential for deploying models in real-world settings. The work has high significance for the medical and neuroscience community.
- It presents a framework that improves diagnostic tools for neurological disorders like AD and ASD, potentially leading to earlier and more accurate diagnoses.
- The authors evaluate their method across two major datasets (ABIDE and ADNI) and compare it with 16 baselines including brain-specific networks, which adds credibility to their results.
- The alignment of identified brain patterns with known neuroscience findings lends additional weight to the framework's interpretability. Also, ablation study demonstrates the needs of each loss types.

**Weaknesses:**

1) The technical contribution of this paper appears to be marginal despite addressing the OOD generalization problem and enhancing interpretability in brain network analysis. While the introduction of an OOD benchmark for brain networks is appreciated, it is unclear if this benchmark adds novel challenges beyond those already present in multi-site datasets like ABIDE and ADNI. Furthermore, many of the technical components, such as the auxiliary losses and discrete sampling strategy, are borrowed from existing work. Although the paper effectively motivates the need for the Graph Information Bottleneck (GIB) framework, the core technical innovations do not extend significantly beyond prior work.

2) One of the primary technical contributions --- feature selection mechanism --- lacks clarity in its formulation. Specifically, the intuition behind $\hat{X}$ derived from the covariance of $\hat{H}$ and the use of the $tanh()$ as activation function is not well explained, leaving readers uncertain about the necessity of these design choices.

3) The definition of the OOD problem itself also raises concerns. Table 2 indicates insignificant performance differences between in-distribution (ID) and OOD scenarios, even with the Empirical Risk Minimization (ERM) baseline, suggesting that the OOD scenario may not be as challenging as claimed. This raises the possibility that the proposed framework performs effectively only under moderate distribution shifts. Additionally, the paper would benefit from comparing the performance of other brain-specific models, such as BrainNetCNN or BrainGNN, under the same OOD conditions to better contextualize the reported improvements.

4) Grammar should be double checked.

**Questions:**

- Please see the weakness above.

- In addition, can you provide an ablation study on the feature selector and structure extractor by evaluating configurations such as $(X', A)$ and $(X, A')$? These results would help to clearly demonstrate the contribution of each module. Additionally, similar to the discussion on edge scores, the node mask should also be examined to strengthen the claim that the proposed method yields clinically relevant results.

- While several GNNs and HPGNN are incorporated into the framework, certain aspects remain unclear. Specifically, what advantage does using HPGNN with multiple layers (hops) offer over simply multiplying the graph Laplacian matrix, especially if the goal is to capture deviations from local patterns? Furthermore, given your assertion that the brain structure matrix A contains noise, why did you choose to retain A rather than use A’ during feature selection?

---

> ### Author Response · Authors · 2024-11-21
> **Response to Reviewer G3Bt [1/2]**
>
> **[W1. Technical Contribution.]**
>
> We need to clarify that the technical contributions of this work lie in both the methodology level in addressing several unique challenges of BrainOOD, as well as the benchmarking level in illustrating the OOD generalization challenge in brain network analysis.
>
> From the benchmark level, while multi-site datasets like ABIDE and ADNI are commonly used, they are not inherently designed for the evaluation of OOD generalization in brain network analysis. Our proposed benchmark introduces a structured OOD scenario by partitioning these datasets based on site-specific differences. This setup systematically evaluates the robustness of models against unseen site distributions, a novel challenge that existing benchmarks do not explicitly address. Furthermore, our results demonstrate that current methods often struggle with such OOD scenarios, validating the need for a specialized benchmark to drive future research in this domain.
>
> From the methodology level,
> -  BrainOOD indeed raises several unique challenges for OOD generalization and interpretability, which are the noisy inputs and the consistency requirements, respectively.
> - To tackle the noises in the brain networks, we propose a novel feature selection strategy as well as a noise filtering strategy to better extract the meaningful information in the brain network inputs.
> - Accommodating with the challenge of noise in brain network analysis, we provide a theoretical discussion on the failures of previous methods in effectively identifying the desired OOD generalizable and interpretable subgraph. Moreover, we also propose to incorporate a reconstruction-based loss to better tackle with influence of the noises.
> - As previous OOD generalization and interpretability methods fail to identify the desired subgraph, we also empirically demonstrate the superior performance of our proposed approach.
>
> While individual components may have roots in existing techniques, their integration into a unified GIB framework specifically for brain network analysis is novel. The emphasis on tackling OOD generalization—a critical but underexplored issue in this field—represents a meaningful technical advancement. Moreover, the interpretability aspect, enabled by feature selection and visualizable subgraphs, offers practical value for clinical applications.
>
> **[W2. Clarification of Feature Selection Mechanism.]** Thank you for pointing this out. We acknowledge that the explanation could be clearer. In Eq. (8), the use of $Tanh()$ is not intended as a typical activation function. Instead, it serves to scale the range of the reconstructed features to align with the input connectivity matrix, which typically has values in the range [-1, 1]. Additionally, the self-multiplication operation $\hat{\boldsymbol{H}} \hat{\boldsymbol{H}}^\mathsf{T}$ is designed to ensure the output exhibits the symmetry property inherent in the connectivity matrix. This operation mimics the structure of the input data, making it easier for the model to capture meaningful patterns during reconstruction. For clarity, we have added this description in Section 4.2 of our revision.
>
> **[W3. Definition of OOD Problem and Comparison with Brain-Specific Models.]** Thank you for your suggestion. To further evaluate how non-OOD methods perform in an OOD scenario, we report the ID and OOD accuracies for GCN and BrainNetCNN. As shown in the following table, while they achieve good performance on the ID set, a significant gap between ID and OOD results is observed, which indicates these methods fail to generalize to the OOD set. It demonstrates the challenge of OOD generalization for brain network datasets. We have added these results and discussions in Section 5.2 of our revision.
>
> |     OOD model      |     ABIDE           |                     |     ADNI            |                      |
> |--------------------|---------------------|---------------------|---------------------|----------------------|
> |                    |     ID_test         |     OOD_test        |     ID_test         |     OOD_test         |
> |     GCN            |     63.69 ± 3.20    |     56.45 ± 5.52    |     59.95 ± 8.20    |     55.32 ± 10.23    |
> |     BrainNetCNN    |     65.50 ± 4.77    |     60.38 ± 7.07    |     62.08 ± 6.81    |     55.02 ± 11.10    |
>
>
> **[W4. Grammar should be double checked.]** Thank you for your feedback. We have thoroughly reviewed and revised the manuscript to address grammatical issues and enhance clarity throughout. We appreciate your attention to detail and believe the revised version now reads more smoothly.

---

> ### Author Response · Authors · 2024-11-21
> **Response to Reviewer G3Bt [2/2]**
>
> **[Q1. Ablation Study on Feature Selector and Structure Extractor.]**  Thank you for the insightful suggestion. To evaluate the contributions of the feature selector and structure extractor, we performed additional experiments as shown in the following table, where “feat” represents using $\mathbf{X}$ or $\mathbf{X}^\prime$ as the feature matrix, while “adj” represents using $\mathbf{A}$ or $\mathbf{A}^\prime$ as the adjacency matrix. The best results are achieved when both modules are used together, confirming the complementary nature of the feature selector and structure extractor. We can also observe that the adjacency matrix generated by the structure extractor contributes more to the improvement of the OOD set. We have included these results and discussions in Section 5.4 of our revision.
>
> |     feat    |     adj    |     ID acc            |     OOD acc           |     acc               |
> |-------------|------------|-----------------------|-----------------------|-----------------------|
> |     $\mathbf{X}$       |     $\mathbf{A}^\prime$     |     63.56   ± 4.40    |     62.26   ± 5.68    |     62.69   ± 3.42    |
> |     $\mathbf{X}^\prime$     |     $\mathbf{A}$      |     63.71   ± 5.97    |     55.40   ± 8.95    |     60.10   ± 3.47    |
> |     $\mathbf{X}^\prime$     |     $\mathbf{A}^\prime$     |    **64.07** ± 4.58      |     **64.81** ± 9.01      |     **63.95** ± 4.65      |
>
> **[Q2.1. Advantages of HPGNN with Multiple Layers]**  The advantage of HPGNN with multiple layers lies in its ability to aggregate information across increasingly distant neighborhoods. Unlike directly multiplying the graph Laplacian matrix, which assumes linear propagation of features, multiple layers in HPGNN enable non-linear transformations at each hop. This allows the model to better capture complex, hierarchical patterns and deviations from local structures, which are particularly relevant in brain networks where functional and structural connectivity often span multiple scales.
>
> **[Q2.2. Retaining $\mathbf{A}$ During Feature Selection.]** Thank you for your question. The output of the feature selector, the masked feature matrix $\mathbf{X}'$, serves as the input to the structure extractor to generate $\mathbf{A}'$. This sequential processing is intentional. We prioritize feature selection first because it allows the structure extractor to work with cleaner and more relevant features, which is essential for learning a high-quality structure matrix. The decision to prioritize structure extraction after feature selection is supported by our ablation study, which demonstrates that the structure plays a more significant role in the final performance. By providing clearer features, we ensure that the structure extractor can generate a more accurate and robust $\mathbf{A}'$, ultimately enhancing the overall model performance.

---

> > ### Comment · Reviewer_G3Bt · 2024-11-26
> >
> > I appreciate the authors for their answers.
> > What I still do not fully understand is which component of this work directly tackles the OOD problem. Is it the GIB and this work brings more expressiveness to the graph representation of functional networks to let GIB handle the OOD?

---

> > > ### Author Response · Authors · 2024-11-26
> > > **Further clarification on how BrainOOD tackles OOD problem**
> > >
> > > Dear Reviewer G3Bt,
> > >
> > > Thank you for your follow-up question to our responses. BrainOOD's main rationale for tackling the OOD generalization problem is the GIB, which extracts a minimal sufficient subgraph of the input graph that preserves the causal relations under a certain data generation process[1,2].
> > >
> > > The components proposed in BrainOOD include:
> > > - *Feature selector*: Reducing the noises in the brain network inputs, which indirectly contributes to the OOD problem;
> > > - *Reconstruction of the graph representation*: Improving the expressiveness of the graph representation to avoid the failure of GIB as shown in Theorem 4.1, which contributes to resolving the OOD problem most directly;
> > > - *Alignment of substructure selection*: Ensuring the consistency of the selected substructure of the brain network across different samples, which contributes to the interpretability of the brain network analysis;
> > >
> > > Please let us know if the aforementioned explanation clarifies your remaining concerns. Otherwise, we'd like to provide more details! Thank you so much!
> > >
> > >
> > > **References**
> > >
> > > [1] Interpretable and generalizable graph learning via stochastic attention mechanism, ICML'22.
> > >
> > > [2] Learning causally invariant representations for out-of-distribution generalization on graphs, NeurIPS'22.

---

> > > > ### Author Response · Authors · 2024-12-03
> > > > **[Gentle Reminder] Discussion period is closing soon**
> > > >
> > > > Dear Reviewer G3Bt,
> > > >
> > > > Thank you again for your time and valuable suggestions on our work! We understand you are busy. As the discussion period is closing soon, could you please take a look of our response above and let us know if our explanation addresses your concerns? We are more than happy to provide more details if it does not. And we would sincerely appreciate it if you could jointly consider our responses above when making the final evaluation of our work.
> > > >
> > > > Sincerely,
> > > >
> > > > Authors

---

### Official Review · Reviewer_wZYM · 2024-11-04

**Soundness:** 2
**Presentation:** 2
**Contribution:** 2
**Rating:** 6
**Confidence:** 4

**Summary:**

This work addresses the out-of-distribution (OOD) problem in brain network analysis. It introduces a framework called BrainOOD, which consists of a feature selector and a structure extractor. By filtering out noisy nodes and edges and enforcing the model to consistently select the same connections across all brain networks within each batch, the proposed method achieves strong performance on the ABIDE and ADNI datasets. Additionally, visualization results are provided to illustrate the method’s effectiveness.

**Strengths:**

- **Originality**: This paper demonstrates a notable level of novelty, particularly in its combined approach of selecting critical node features and graph structures, along with the batch-level loss designed to identify key discriminative connections.
- **Quality**: The methodology is thoroughly evaluated through comparisons with 16 existing methods across two datasets (ABIDE and ADNI), effectively highlighting its effectiveness and efficiency.
- **Significance**: This research provides valuable insights into addressing the OOD problem in brain network analysis, contributing meaningfully to advancements in neuroscience.

**Weaknesses:**

- **Contribution of the Benchmark**

**The claim of introducing the first benchmark seems somewhat overstated.** The ABIDE and ADNI datasets have been long established in brain network analysis and are widely used for evaluating brain disorder diagnosis models. Simply partitioning these datasets to create an OOD scenario may not constitute a significant contribution.

- **Alignment of Motivation, Method, and Analysis**

The motivation of this work is to address the OOD generalization problem. However, **it is not clearly explained how the proposed method specifically tackles this issue**. While reducing noisy nodes and structures could indeed improve brain disorder diagnosis performance, the methodology and interpretive analysis lack clarity on how this approach mitigates the OOD generalization problem. For instance, visualizing the top 10 connections with the highest scores on both the ABIDE ID and ABIDE OOD sets could help demonstrate the method’s generalizability more effectively.

- **Paper Organization**

The organization of the paper could be improved for clarity. **It may not be necessary to dedicate extensive sections to GNN and brain network fundamentals.** Additionally, placing the related work section directly after the introduction or immediately before the conclusion could improve the flow and readability.

**Questions:**

1. How do you balance the four losses in the proposed method? Given the numerous modules and hyperparameters involved, does training the model from scratch carry a high risk of overfitting?
2. Considering the frequent occurrence of the OOD generalization problem in brain network analysis, how could the proposed method be adapted or transferred to other models?
3. Since the performance of fMRI-derived brain networks on the ADNI dataset is lower than that of structural MRI, do you believe it is appropriate or necessary to use it as a benchmark for the OOD generalization problem?

---

> ### Author Response · Authors · 2024-11-21
> **Response to Reviewer wZYM [1/2]**
>
> **[W1. Contribution of the Benchmark.]** We appreciate the reviewer’s comment and would like to clarify our contribution regarding the benchmark setup. While it is true that the ABIDE and ADNI datasets are widely used in brain network analysis for evaluating brain disorder diagnosis models, existing studies typically utilize these datasets without explicitly addressing the challenge of **distribution shifts** across different acquisition sites. In most prior work, the focus has been on in-distribution (ID) evaluation, which assumes a uniform data distribution without taking into account the inherent domain shifts present across different data collection sites. In our work, we go beyond the conventional usage of these datasets by creating a specific **OOD benchmark scenario** that simulates real-world conditions where models encounter data from unseen sites during testing. This setup is motivated by the well-known issue of site-specific biases in neuroimaging data, which often lead to significant distribution shifts.
> - To address this, we carefully partition the ABIDE and ADNI datasets across multiple folds, ensuring that each fold contains test sets with both ID and OOD subjects. By selecting the smallest/largest sites as  OOD test sets based on the dataset characteristics (ABIDE: small sites; ADNI: large sites), we ensure a diverse and realistic OOD evaluation protocol.
> - Our proposed benchmark is the first to systematically evaluate **OOD generalization** on brain network datasets with a focus on addressing site-specific variability, which is a critical challenge in clinical applications.
> - The experimental results show that existing OOD methods fail to generalize well in this scenario, highlighting the necessity of specifically designed OOD algorithms for brain network data. We have revised the manuscript to further clarify this point in the introduction. We hope this explanation better conveys the significance of our benchmark in advancing OOD research for brain network analysis.
>
>
> **[W2. Alignment of Motivation, Method, and Analysis.]**
>
> The rationale of our proposed method to resolve the OOD generalization relies on the concept of graph information bottleneck, which aims to find a minimal sufficient subgraph to predict the label. Previous works show that predictions based merely on the minimal sufficient subgraph are able to generalize to OOD graphs [1,2]. Finding the minimal sufficient subgraph is essentially to remove the noisy and spuriously correlated substructures from the original input data.
>
> However, the noisy nature of fMRI data and the limited encoding capacity of GNNs raise unique challenges in implementing the graph information bottleneck, as shown in our Theorem 4.1. Therefore, we propose several strategies to resolve the challenges and better identify the desired subgraphs.
>
> For the analysis of the method’s generalizability, current visualization can provide some evidence. For the analysis of the method’s generalizability, current visualization can provide some evidence. As shown in Figure 3, the OOD model (top left) and the ID model (bottom right) highlight some common connections, particularly within regions such as the VIS, SMN, VAN, and DMN networks. While the specific scores differ, the overall patterns in these regions remain consistent. This observation suggests that BrainOOD captures critical features that are robust across both ID and OOD scenarios, indicating its strong generalization ability.
>
> **[W3. Paper Organization.]** We included the background sections on GNNs and brain networks to cater to a broader audience, as our work intersects multiple domains (machine learning and neuroscience). However, we understand that this level of detail may not be necessary for readers already familiar with these topics. In response to your suggestion, we have streamlined these sections to focus on the most relevant aspects and reduce redundant explanations in our revision. Besides, we have moved the related work section after the experiment section in the revision to enhance the clarity and readability of the paper.

---

> ### Author Response · Authors · 2024-11-21
> **Response to Reviewer wZYM [2/2]**
>
> **[Q1. ​​Balancing the Loss Functions and Risk of Overfitting.]** Balancing multiple loss functions is indeed a critical aspect of our proposed method. We employ a series of trade-off hyperparameters to combine the four auxiliary losses. The weights are treated as hyperparameters and tuned using grid search on the validation set.
>
> Regarding the risk of overfitting, we acknowledge that the complexity of the model and the inclusion of multiple modules may increase the likelihood of overfitting, particularly on small datasets. To mitigate this risk, we adopt several strategies:
> - (1) Early Stopping: We monitor the validation loss and stop training when there is no improvement over a predefined number of epochs.
> - (2) Regularization Techniques: We apply dropout and L2 regularization to reduce the model’s capacity and encourage generalization.
> - (3) Cross-Validation: We use 10-fold cross-validation, ensuring that our results are robust and not dependent on a specific split.
>
> These implementation details are included in Appendix D.2.
>
> **[Q2. Adaptability and Transferability to Other Models.]** We appreciate the reviewer’s interest in the adaptability of our method. The BrainOOD framework is designed to be modular, making it straightforward to incorporate with various GNN backbones. Apart from the GIN backbone used in the main results, we also integrated BrainOOD with the GCN backbone, which shows consistent performance gains. This discussion is included in Appendix E.2. For broader applicability beyond the specific functional brain networks evaluated, we believe that the transferability of BrainOOD can further benefit models that encounter domain shifts in other types of neuroimaging data, such as DTI or EEG.
>
> **[Q3. Appropriateness of Using fMRI-Derived Brain Networks for OOD Generalization.]** Our research is based on the brain network datasets proposed by [3], which focus on fMRI. We agree that evaluating our proposed method on brain networks that are constructed from various types of neuroimaging would enhance its robustness. However, our main focus is still developing an algorithm that could improve the OOD generalization ability of GNNs, rather than selecting the most appropriate modality for brian network construction. Moreover,  fMRI-derived brain networks often exhibit greater variability across different sites and acquisition conditions, making them an ideal candidate for assessing the robustness of OOD methods.
>
> [1]  Interpretable and generalizable graph learning via stochastic attention mechanism. ICML 2022.
>
> [2] Learning causally invariant representations for out-of-distribution generalization on graphs. NeurIPS 2022.
>
> [3] Data-driven network neuroscience: On data collection and benchmark. NeurIPS 2023

---

### Author Response · Authors · 2024-11-25
**General Response**

We sincerely appreciate the reviewers' constructive feedback and positive remarks on our paper. We are grateful for the recognition of our work as **novel** (@wZYM, GyEy), **thoroughly evaluated** (@wZYM, G3Bt), and having **high significance for the medical and neuroscience community** (@wZYM, G3Bt), while also **offering new insights to the field of neuroscience** (@G3Bt).

In this rebuttal, we have thoroughly addressed the reviewers' main concerns and provided additional experiments and clarifications to strengthen our work. Revisions in the manuscript are highlighted in blue for easy reference. We remain open to further discussions to resolve any remaining concerns or address additional questions.

---

### Meta-Review · Area_Chair_7Yfm · 2024-12-19

**Metareview:**

This submission contributes an analysis method for brain networks to help OOD generalization, via feature selection and structure extraction with a graph information bottleneck. It led to interest and discussion with the reviewers, which appreciated the prediction-gain brought by the method. The analysis of how specifically the method brings this robustness to distribution shift was however perceived as light.

**Additional Comments On Reviewer Discussion:**

There was a good discussion with much back and forth between authors and reviewers. The discussion led to improving the manuscript, notably with new baselines.

---

### Decision · Program_Chairs · 2025-01-22

Accept (Poster)